# Kenny mediates selective autophagic degradation of the IKK complex to control innate immune responses

Radu Tusco [1], Anne-Claire Jacomin [1], Ashish Jain [2,3,4], Bridget S. Penman[1], Kenneth Bowitz Larsen[2], Terje Johansen[2] & Ioannis P. Nezis[1]

Selective autophagy is a catabolic process with which cellular material is specifically targeted for degradation by lysosomes. The function of selective autophagic degradation of self-components in the regulation of innate immunity is still unclear. Here we show that *Drosophila* Kenny, the homolog of mammalian IKKγ, is a selective autophagy receptor that mediates the degradation of the IκB kinase complex. Selective autophagic degradation of the IκB kinase complex prevents constitutive activation of the immune deficiency pathway in response to commensal microbiota. We show that autophagy-deficient flies have a systemic innate immune response that promotes a hyperplasia phenotype in the midgut. Remarkably, human IKKγ does not interact with mammalian Atg8-family proteins. Using a mathematical model, we suggest mechanisms by which pathogen selection might have driven the loss of LIR motif functionality during evolution. Our results suggest that there may have been an autophagy-related switch during the evolution of the IKKγ proteins in metazoans.

[1] School of Life Sciences, University of Warwick, CV4 7AL Coventry, UK. [2] Molecular Cancer Research Group, Institute of Medical Biology, University of Tromsø – The Arctic University of Norway, 9037 Tromsø, Norway. [3] Department of Molecular Cell Biology, Institute for Cancer Research, Oslo University Hospital, Montebello, N-0379 Oslo, Norway. [4] Centre for Cancer Biomedicine, Faculty of Medicine, University of Oslo, Montebello, N-0379 Oslo, Norway. Radu Tusco and Anne-Claire Jacomin contributed equally to this work Correspondence and requests for materials should be addressed to I.P.N. (email: I.Nezis@warwick.ac.uk)

Innate immunity constitutes one of host's first defense against pathogen invasion and relies on inflammatory responses controlled by the nuclear factor-kappa B (NF-κB) family transcription factors[1]. These NF-κB associated pathways regulate the expression of immune and stress response genes aiming at clearing pathogens. However, improper activation of these pathways can cause pathologies such as inflammation and cancer. Thus, to maintain tissue homeostasis and organism survival, the activation of NF-κB associated pathways must be tightly regulated[1–3]. *Drosophila* innate immunity is controlled by two major signaling pathways: the immune deficiency (IMD) and Toll pathways. The activation of either pathway results in the production of antimicrobial peptides (AMPs) that neutralize the bacterial load[4–6].

Autophagy is an evolutionarily conserved process, by which cells degrade their own cellular material. Autophagy serves as a cellular response to nutrient starvation, and for the removal of aggregated proteins, lipids, damaged organelles, and invading bacteria and viruses[2,7–13]. Autophagy is subdivided into three types (macroautophagy, microautophagy, and chaperone-mediated autophagy) of which macroautophagy (hereafter referred to as "autophagy") is the most well-studied. During autophagy there is sequestration of cellular material into double-membrane vesicles called autophagosomes. The autophagosomes eventually fuse with lysosomes where the sequestered cargos are degraded by lysosomal hydrolases[14]. The sequestration and degradation of cytoplasmic material by autophagy can be selective through the action of specific receptor proteins[9,15,16]. Selective autophagy receptors usually contain a short motif, termed LC3-interacting region (LIR) or Atg8-interacting motif (AIM), that is necessary for their interaction with the autophagosomal membrane protein microtubule-associated protein 1 light chain 3 (Atg8/LC3)[16–18]. Atg8/LC3 is required for the proper formation and maturation of autophagosomes, and serves as a central scaffold protein for selective autophagy receptors and cargos[19]. The implication of autophagy in the degradation of intracellular pathogens has been widely described[13]. However, the molecular mechanisms that link selective autophagy of cellular self-components and innate immunity remains largely unexplored.

Using an in silico approach to identify *Drosophila* proteins that interact with Atg8/LC3 proteins, here we show that Kenny, the *Drosophila* homolog of IKKγ/NEMO, is an Atg8a-interacting protein, and is required for the selective autophagic degradation of the IκB kinase (IKK) complex to prevent the constitutive activation of the IMD pathway by commensal bacteria. We additionally show that mammalian IKKγ/NEMO lacks a functional LIR motif. Finally, we present a mathematical model which explores the evolution of the LIR motif during host interaction with pathogens and commensal microbiota. Overall, our results describe a molecular mechanism of selective autophagy in *Drosophila* innate immunity, and suggest that pathogen-related selective pressures may have induced the loss of a functional LIR motif in mammalian IKKγ through evolution.

## Results

**Kenny is an Atg8a-interacting protein**. In order to identify LC3-interacting region (LIR) motif-containing proteins in *Drosophila*, we screened the *Drosophila* proteome for the presence of LIR motifs using the iLIR software that we have developed[20,21]. We focused on proteins that also contained ubiquitin-binding domains[22]. One of our hits was the UBAN domain-containing protein Kenny (CG16910) that has a predicted LIR motif at its N terminus (Fig. 1)[20,21]. Kenny is the *Drosophila* homolog of mammalian IKKγ/NEMO and has been shown to control innate immunity in *Drosophila* through the

immune deficiency (IMD) pathway[23,24]. We found that in vitro translated Kenny ([35S]-Myc-Kenny) binds to recombinant GST-Atg8a, the *Drosophila* homolog of Atg8/LC3 (Fig. 2a). This interaction was abolished by alanine substitutions of the aromatic and hydrophobic residues (F7A and L10A) of the LIR motif of Kenny (Fig. 2a). This observation was further verified in *Drosophila* S2R+ cells. eGFP-Atg8a was co-precipitated with wild-type 3XFLAG-Kenny while there was a 62% decrease in the interaction with its LIR motif mutant counterpart (Fig. 2b). Atg8-family proteins are able to interact with proteins that contain canonical LIR motifs via their LIR docking site (LDS)[15,17,25]. We observed that a K48A/Y49A LDS mutant of Atg8a was unable to interact with Kenny (Fig. 2a). In order to investigate the functional relevance of the LIR motif in Kenny, we constructed transgenic flies carrying either UAS-GFP-Kenny[WT] or UAS-GFP-Kenny[F7A/L10A] and expressed them in the larval fat body together with the autophagic marker mCherry-Atg8a[26]. We observed that wild-type GFP-Kenny, but to a lesser extent its LIR motif-mutated form, colocalized with mCherry-Atg8a-positive autophagosomes after starvation-induced autophagy (Fig. 2c–e). We also observed that wild-type GFP-Kenny, but not its LIR motif-mutated counterpart, co-localized with active lysosomes as monitored by Cathepsin-L staining (Supplementary Fig. 1). Taken together these results indicate that Kenny is an Atg8a-interacting protein and that this interaction is LIR motif-dependent.

**Kenny is selectively degraded by autophagy**. Given the observed interaction between Kenny and Atg8a, we examined whether Kenny is degraded by autophagy. Western blot analysis of whole body fly lysates showed a robust accumulation of Kenny in *Atg8a* and *Atg7* mutants compared to wild-type flies (Fig. 3a–c). Accumulation of Kenny protein was found to be a post-translational event since mRNA levels for Kenny were found to be similar between wild-type and autophagy mutant flies (Supplementary Fig. 2). Immunofluorescence analysis of endogenous Kenny in various tissues revealed that Kenny accumulates and forms cytoplasmic aggregates that co-localize with Ref(2)P and ubiquitinated proteins in autophagy mutant fly tissues, such as adult brain, fat body and midgut, compared to wild-type flies (Fig. 3d–l, Supplementary Fig. 3). This observation was further reinforced by mosaic analysis which clearly showed an accumulation of endogenous Kenny in autophagy-depleted (*Atg5*-RNAi and *Atg8a*-RNAi) and autophagy mutant (*Atg1* and *Atg13*) clonal cells but not in their wild-type neighbors (Fig. 4). Furthermore, expression of mCherry-eYFP-Kenny[WT] and mCherry-eYFP-Kenny[F7A/L10A] in HeLa cells showed that Kenny was degraded in the lysosomes in a manner dependent both on an intact LIR motif and co-expression of Atg8a (Fig. 5).

The ubiquitin-proteasome system (UPS) constitutes another major intracellular proteolytic system in eukaryotes[27]. To explore the contribution of the UPS in the degradation of Kenny, adult flies were fed with food supplemented with the proteasome inhibitor Bortezomib or vehicle only. We found that there was a modest (compared to the accumulation observed in autophagy mutants) accumulation of Kenny upon proteasomal inhibition, suggesting that Kenny is predominantly degraded by autophagy (Supplementary Fig. 4). Taken together these results show that Kenny is preferentially degraded by autophagy in a LIR-dependent and Atg8a-dependent manner in basal conditions.

**Autophagy mutant flies accumulate phosphorylated Kenny.** We observed that Kenny protein resolved into two bands in lysates from *Atg8a* and *Atg7* mutant flies while a single band was observed in wild-type flies (Fig. 3b). The presence of a doublet of

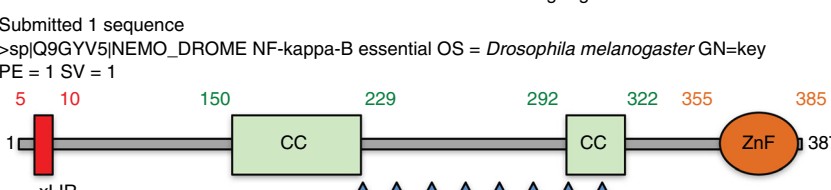

**Fig. 1** Kenny is a UBAN domain-containing protein with a xLIR motif. Screenshot of the result window from the iLIR server (http://ilir.warwick.ac.uk/) for the protein sequence of Kenny (UniProt Q9GYV5). xLIR: relaxed LIR motif; WxxL: conventional LIR motif; CC: coiled coil; UBAN: ubiquitin binding in ABIN and NEMO domain; ZnF: Zinc finger motif; PSSM: position-specific scoring matrix

bands can be associated with post-translational modifications, such as phosphorylation or ubiquitination, which have been shown to be involved in mammalian NEMO regulation[28,29]. We investigated the possibility of phosphorylation as a cause of appearance of Kenny doublet bands by treating lysates from autophagy-deficient flies with either calf intestinal phosphatase (CIP) or lamba phosphatase (λPP). In each case, we noticed that the doublet of bands converted into a single band corresponding to the lower molecular weight band (Supplementary Fig. 5a). To test whether ubiquitinated Kenny accumulates in autophagy-deficient flies, GFP-Kenny was expressed in wild-type or *Atg8a* mutant flies and consequently immunoprecipitated from denaturated fly lysates. No ubiquitination pattern could be detected using a pan-ubiquitin chain antibody (Supplementary Fig. 5b). These results show that Kenny is post-translationally modified by phosphorylation and is accumulated in its phosphorylated form in autophagy mutants.

**Kenny is required for the autophagosomal degradation of ird5.** Kenny is a component of the IκB kinase (IKK) complex which consists of IKKβ/ird5 and IKKγ/Kenny, and is crucial for nuclear translocation of transcription factor Relish and induction of the expression of antimicrobial peptide (AMP) genes, including *Diptericin* (*Dpt*)[23,30–33]. Using *UAS-mCherry-ird5* transgenic flies, we observed that mCherry-ird5 is diffused in the cytoplasm of fat body cells in fed conditions (Fig. 6a). However, mCherry-ird5 displayed a dotted localization after starvation, and co-localized with Atg8a and Cathepsin-L (Fig. 6b–d). Using a *UAS-GFP-mCherry-ird5* transgenic line to distinguish acidic and non-acidic compartmentalization of ird5, we observed an accumulation of mCherry-only (acidic) ird5 puncta in starved conditions (Fig. 6e, f). mCherry-only puncta exclusively co-localized with the lysosomal marker Cathepsin-L (Fig. 6g). These results suggest that ird5 is targeted to the autophagosomes to undergo lysosomal degradation. This was further confirmed by the accumulation of mCherry-ird5 when larvae were fed with the lysosomal inhibitor chloroquine (Fig. 6h). Interestingly, we observed that adult flies fed with Bortezomib exhibited a reduction of mCherry-ird5 protein levels (Supplementary Fig. 6). This observation reinforces

the hypothesis that ird5 is degraded by autophagy as impairment of the UPS enhances autophagic degradation[34]. We then expressed mCherry-ird5 along with GFP-Kenny wild-type or its LIR-defective counterpart. We observed that the LIR motif in Kenny was not required for mCherry-ird5 and GFP-Kenny interaction (Fig. 7a, b and Supplementary Fig. 7). An in vitro GST pull-down experiment confirmed the LIR-independent direct interaction between ird5 and Kenny (Fig. 7c)[24]. In order to examine whether the targeting of the whole IKK complex to the autolysosome is dependent on Kenny's LIR motif, we probed fat bodies from starved larvae with an anti-Cathepsin-L antibody. While we observed a clear co-localization of the three proteins when GFP-Kenny[WT] is expressed (Fig. 7d), the co-localization between mCherry-ird5 and the lysosomes was lost when GFP-Kenny[F7A/L10A] is expressed (Fig. 7e). Additionally, we observed that starvation-induced mCherry-ird5 accumulation is more prominent upon *kenny* RNAi-mediated knockdown (Supplementary Fig. 8). Finally, we observed that the size of the starvation-induced mCherry-ird5 puncta was reduced when *key* RNAi was co-expressed, compared to control RNAi (Fig. 7f–h), suggesting less autophago-lysosomal targeting. Collectively, these results show that Kenny is a selective autophagy receptor for the degradation of the IKK complex by autophagy.

**The IMD pathway is controlled by selective autophagy.** In order to examine the physiological significance of IKK complex accumulation in autophagy-depleted flies, we analyzed the expression level of the IMD target gene *Dpt*. We observed that there was a significant systemic upregulation of *Dpt* in *Atg8a* and *Atg7* mutant flies (Fig. 8a, one-way ANOVA test ***$P < 0.001$, ****$P < 0.0001$) that can be partially rescued by re-expressing mCherry-Atg8a in the fat body (Supplementary Fig. 9). Furthermore, the upregulation of *Dpt* was accompanied with nuclear localization of Relish in fat body cells (Fig. 8b–d). However, although we could detect nuclear translocation of Relish in autophagy mutant gut cells, no significant upregulation of the *Dpt* gene expression could be detected in isolated guts (Supplementary Fig. 10, one-way ANOVA, $P > 0.05$), suggesting

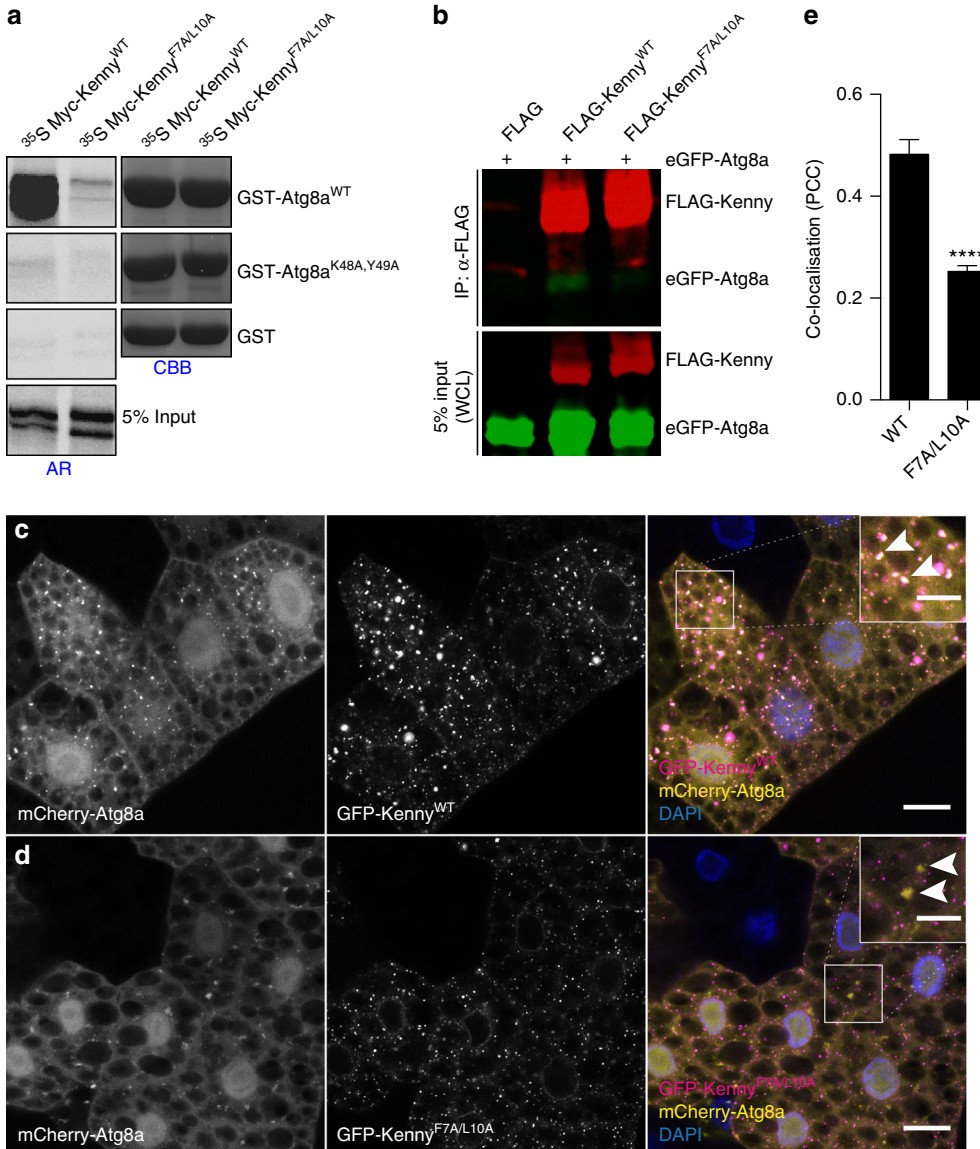

**Fig. 2** Kenny interacts with Atg8a in a LIR-dependent manner. **a** GST-pull-down assay between GST-tagged Atg8a-WT or -LDS mutant (K48A, Y49A), and radiolabelled myc-Kenny- WT or -LIR mutant (F7A/L10A). **b** Anti-Flag immunoprecipitation (IP) between eGFP-Atg8a and Flag-Kenny-WT or –F7A/L10A from whole S2R+ cell lysates (WCL) and subjected to SDS-PAGE. **c, d** Confocal images of fat body from starved larvae clonally expressing mCherry-Atg8a (yellow) and GFP-Kenny-WT or -F7A/L10A mutant (magenta). Scale bars are 20 μm (10 μm in the insets). Arrowheads in insets show co-localization of mCherry-Atg8a and GFP-Kenny-WT in **c** and no co-localization in **d**. **e** Quantification of the co-localization of mCherry-Atg8a and GFP-Kenny signals using the Pearson's correlation coefficient. Bar chart shows means ± s.d. Statistical significance was determined using two-tailed Student's *t*-test, ****$P < 0.001$

that the systemic deregulation of the IMD pathway is predominantly controlled by the fat body. To assess whether the deregulation of the IMD pathway in autophagy-deficient flies is related to the accumulation of Kenny protein, we created *Atg8a; kenny* double mutant flies. We observed that the absence of *kenny* expression in autophagy-deficient flies abrogates the constitutive activation of *Dpt* observed in *Atg8a* mutant flies (Fig. 8e). Together, these results show that the upregulation of the IMD pathway when autophagy is blocked depends on Kenny.

In order to test whether the presence of commensal bacteria is responsible for the upregulation of *Dpt* in autophagy-deficient flies, we examined its expression in flies reared in germ-free (axenic) conditions. Interestingly, no upregulation of *Dpt* was observed in axenic *Atg8a* and *Atg7*-depleted flies compared to their conventionally reared siblings (Fig. 8a). Together these

results indicate that selective autophagic degradation of IKK complex prevents constitutive activation of the IMD pathway in response to commensal microbiota.

**Autophagy mutant flies exhibit hyperplasia in the gut.** In order to examine the physiological effect of systemic upregulation of the IMD pathway in conventionally reared autophagy-deficient flies, we examined cell proliferation rates by immunostaining posterior midguts for phospho-Histone H3 (pH3), a specific marker for mitotic cells. We observed that *Atg8a* and *Atg7* mutant flies exhibited higher numbers of pH3-positive cells compared to wild-type control flies (Fig. 8f–h, l). Interestingly this phenotype was not observed in guts from axenic *Atg8a* and *Atg7* mutant flies (Fig. 8i–k), suggesting that persistent deregulation of the IMD

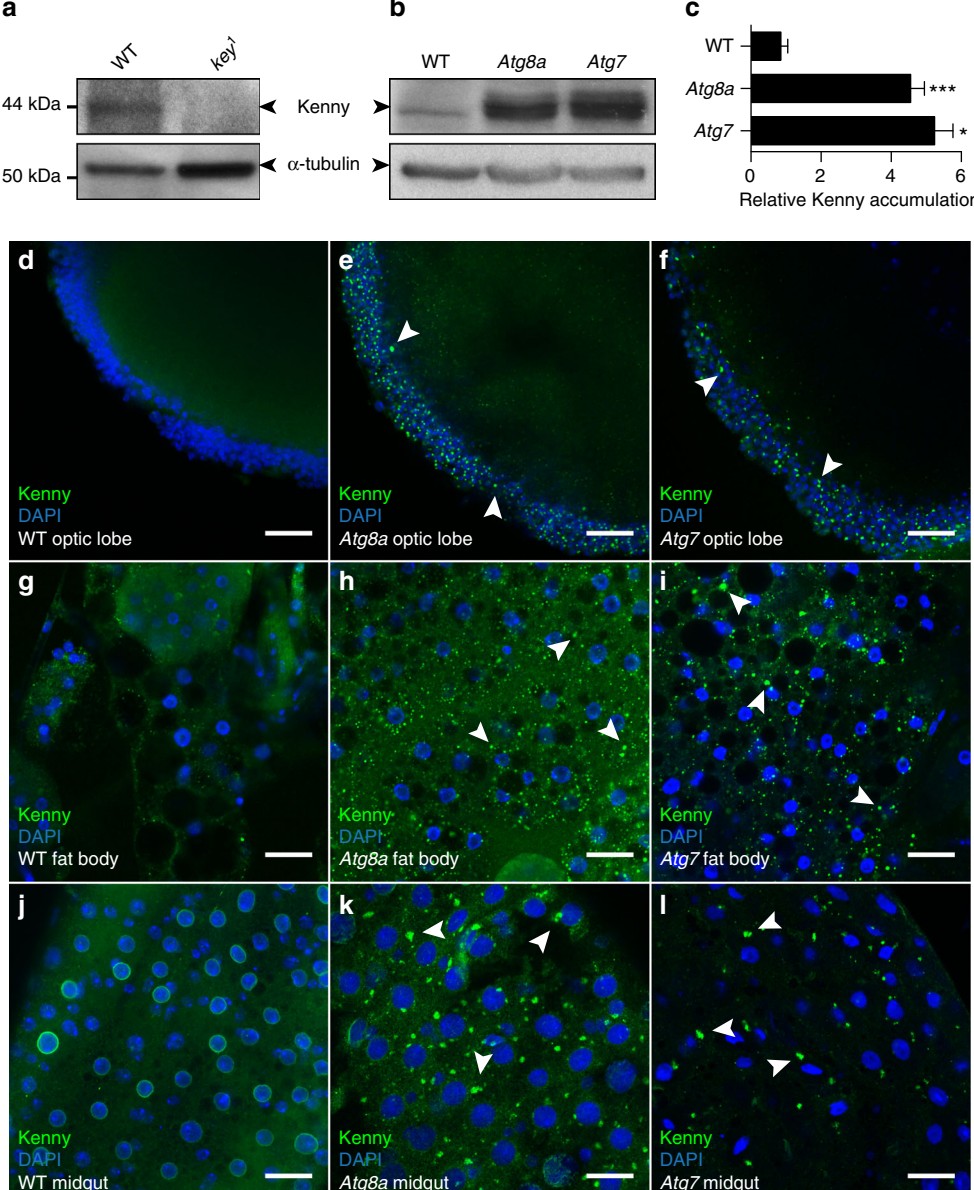

**Fig. 3** Endogenous Kenny protein accumulates in *Atg8a* and *Atg7* mutant flies. **a**, **b** Full body lysates from wild-type (WT) and *kenny* mutant (*key*[1]) flies **a** or *Atg8a* and *Atg7* mutant flies **b** were subjected to SDS-PAGE and immunoblotting for Kenny. Tubulin was used as loading control. **c** Quantification of the quantity of Kenny protein normalized to tubulin. Bar chart shows means ± s.d. Statistical significance was determined using one-way ANOVA, *$P < 0.05$, ***$P < 0.001$. **d–l** Confocal images from adult brains **d–f**, fat bodies **g–i** and midguts **j–l** from WT **d**, **g**, **j**, *Atg8a* **e**, **h**, **k** and *Atg7* **f**, **i**, **l** mutant flies stained for Kenny (green) and DNA (blue). Arrowheads show some Kenny aggregates. Scale bars are 20 µm

pathway as a result of stimulation from commensal bacteria induces a hyperplasia phenotype in autophagy mutant flies. Additionally, we observed that there is no increased pH3 staining in *Atg8a;kenny* mutants (compared to *Atg8a* mutants), indicating that Kenny-mediated upregulation of AMPs is causative of gut hyperplasia in autophagy mutant flies (Supplementary Fig. 11). Finally, we performed clonal expression of GFP-Kenny[WT] and GFP-Kenny[LIR] mutant in adult midgut. We observed that pH3-positive cells were present only in cells expressing GFP-Kenny[LIR] mutant and not in cells expressing GFP-Kenny[WT] (Supplementary Fig. 12).

To understand the physiological relevance of intestinal dysplasia caused by autophagy depletion, we analyzed gut function in wild-type and autophagy mutant flies. Feeding flies with food containing blue dye revealed that flies with dysplastic

guts defecated significantly less (Fig. 8m, one-way ANOVA, *$P < 0.05$, **$P < 0.01$, ***$P < 0.001$, ****$P < 0.0001$), indicating increased retention of ingested food. In addition, we observed that rearing *Atg8a* mutant flies in axenic conditions increased their median lifespan (Supplementary Fig. 13). Together these results show that an uncontrolled systemic immune response to commensal bacteria in *Drosophila* promotes a dysplastic dysfunctional gut that contributes to organismal death.

**Mammalian IKKγ has lost its LIR motif during evolution.** In order to explore the conservation of Kenny LIR motif in arthropods, we used BLAST analysis. We found that Kenny's LIR motif is conserved in other *Drosophila* species, in the common house fly (*Musca domestica*), mosquito (*Aedes aegypti*), butterfly

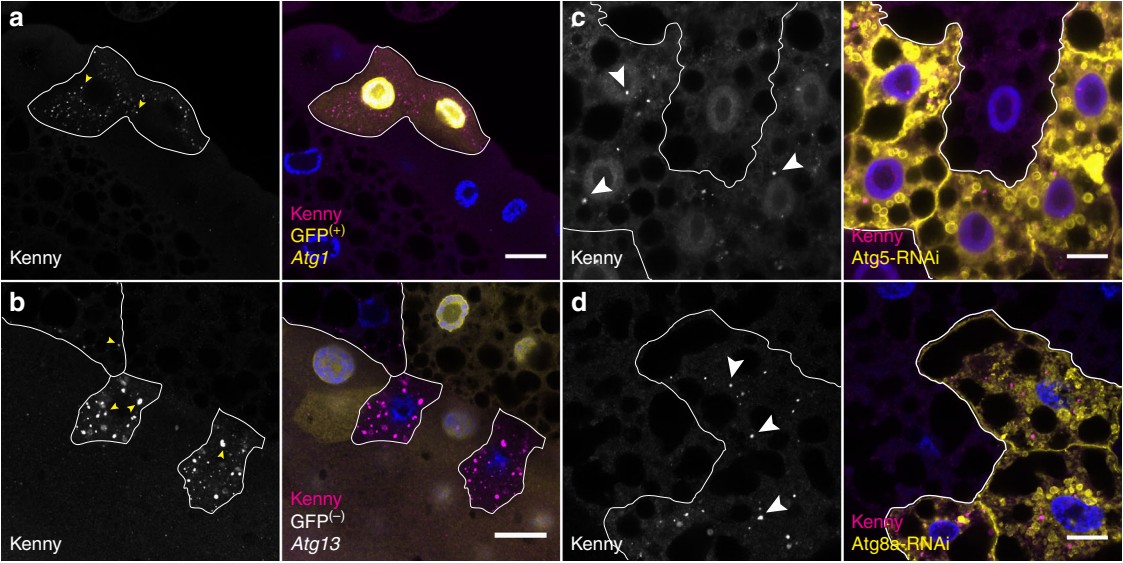

**Fig. 4** Kenny protein accumulates in cells lacking different components of the core autophagy machinery. Confocal images from fat body or salivary glands clonally lacking the expression of key components of the core autophagy machinery, and stained for endogenous Kenny (magenta) protein and Hoechst (blue). **a** *Atg1* null mutant cells expressing nuclear GFP (yellow) were generated by MARCM. **b** *Atg13* null mutant cells, lacking the expression of nuclear GFP (yellow), were generated by FRT/FLP recombination. **c**, **d** RNAi silencing of *Atg5* **c** and *Atg8a* **d** in cells expressing mCD8-GFP (yellow) were performed using the FLPout system. Arrowheads show some Kenny aggregates. Scale bars are 20 μm

(*Papilio xuthus*), and silk moth (*Bombyx mori*) suggesting that there is conservation between Diptera and Lepidoptera (Supplementary Fig. 14). To examine whether Kenny's LIR motif is evolutionarily conserved in mammals, we tested the interaction of its human homolog IKKγ/NEMO with mammalian Atg8-family proteins. Interestingly, we found that IKKγ/NEMO does not interact with any of mammalian Atg8-family proteins (LC3A/B/C, GABARAP/L1/L2), which is consistent with the fact that we could not identify any predicted functional LIR motif in its sequence (Supplementary Fig. 15). In addition, using *UAS-GFP-NEMO* transgenic flies, we observed that unlike GFP-Kenny, human NEMO was unable to co-localize with mCherry-Atg8a-positive autophagosomes (Supplementary Fig. 16).

To understand how the functionality of the LIR motif in IKKγ may have been lost during evolution, we developed a deterministic mathematical model, in which different host types, with and without IKKγ LIR motifs, compete in the presence of a pathogen and a proxy for a member of the microbiota (Fig. 9). Mammalian pathogens have been observed to produce factors which interact directly with NEMO/IKKγ and promote the degradation of NEMO/IKKγ by autophagy[35]. We therefore allowed the pathogen in our model to exist as two variants: one encoding a protein with a functional LIR motif and one without.

Our results have shown that *Drosophila* Kenny/IKKγ can be directly targeted for degradation in a lysosome by interacting with Atg8a via its LIR motif. However, NEMO/IKKγ in mammals has been shown to be ubiquitinated itself and therefore could also interact with selective autophagy receptors via ubiquitin tags[29,36,37]. Mammals are known to possess a greater range of selective autophagy receptors than flies—thus the major mechanism by which mammalian IKKγ is likely to be degraded during infection probably involves ubiquitin tagging of IKKγ followed by autophagy. We therefore considered 4 hypothetical host types within our model: host type 1, in which IKKγ lacks a LIR motif and cannot be tagged for autophagic degradation by ubiquitination; host type 2, in which IKKγ posseses a LIR motif and cannot be tagged for autophagic degradation by ubiquitination; host type 3, in which IKKγ lacks a LIR motif and can be tagged for autophagic degradation by ubiquitination, and host type 4, in

which IKKγ possesses a LIR motif and can be tagged for autophagic degradation by ubiquitination. Type 2 hosts are *Drosophila*-like, and type 3 hosts are human-like.

Full details of the mathematical model and its parameters are found in the Methods section, but the following biological assumptions were applied at all times: (1) A host with neither a LIR motif on IKKγ nor the specific molecular machinery to target IKKγ for autophagic degradation using a ubiquitin tag (host type 1) suffers excess mortality due to the stimulation of the innate immune response by the microbiota. (2) Maintaining a regulatory system capable of appropriately degrading IKKγ using ubiquitination is inherently more costly than regulating IKKγ by LIR induced degradation, due to the proteins required to ubiquitinate IKKγ and the need for appropriate selective autophagy receptors.

To explore the possible evolutionary reasons why the LIR motif may have been lost during mammalian evolution, we considered the following biologically plausible effects: (Effect I) If the host can up or downregulate the degradation of IKKγ by ubiquitin tagging, this may afford greater precision in regulating immunopathology during infection, and result in reduced mortality during infection. (Effect II) As in Effect I, but the advantage of being able to precisely regulate the rate of degradation of IKKγ by ubiquitin tagging may be reduced if a LIR motif is present on IKKγ, constantly signaling it for autophagy. (Effect III) If the pathogen encodes a LIR motif, this may enhance the degradation of IKKγ during infection and attenuate the innate immune response. The consequences of this (both of which could be advantageous to the pathogen) may be (IIIa) a slower recovery rate from infection, or (IIIb) reduced host mortality during infection. We assumed that the pathogen is only able to manipulate the host in this way if host IKKγ lacks a LIR motif (host types 1 and 3).

Figure 9 illustrates the consequences for host and pathogen of applying parameter values capturing these different effects. In the absence of Effects I or II, or any pathogen LIR Effect (IIIa or IIIb), host type 2 dominates, in the presence of a non-LIR encoding pathogen (Fig. 9a, top left hand panel). This is a *Drosophila*-like scenario (Fig. 9b). Host type 1 cannot succeed in this scenario because it suffers excess mortality from the stimulation of innate

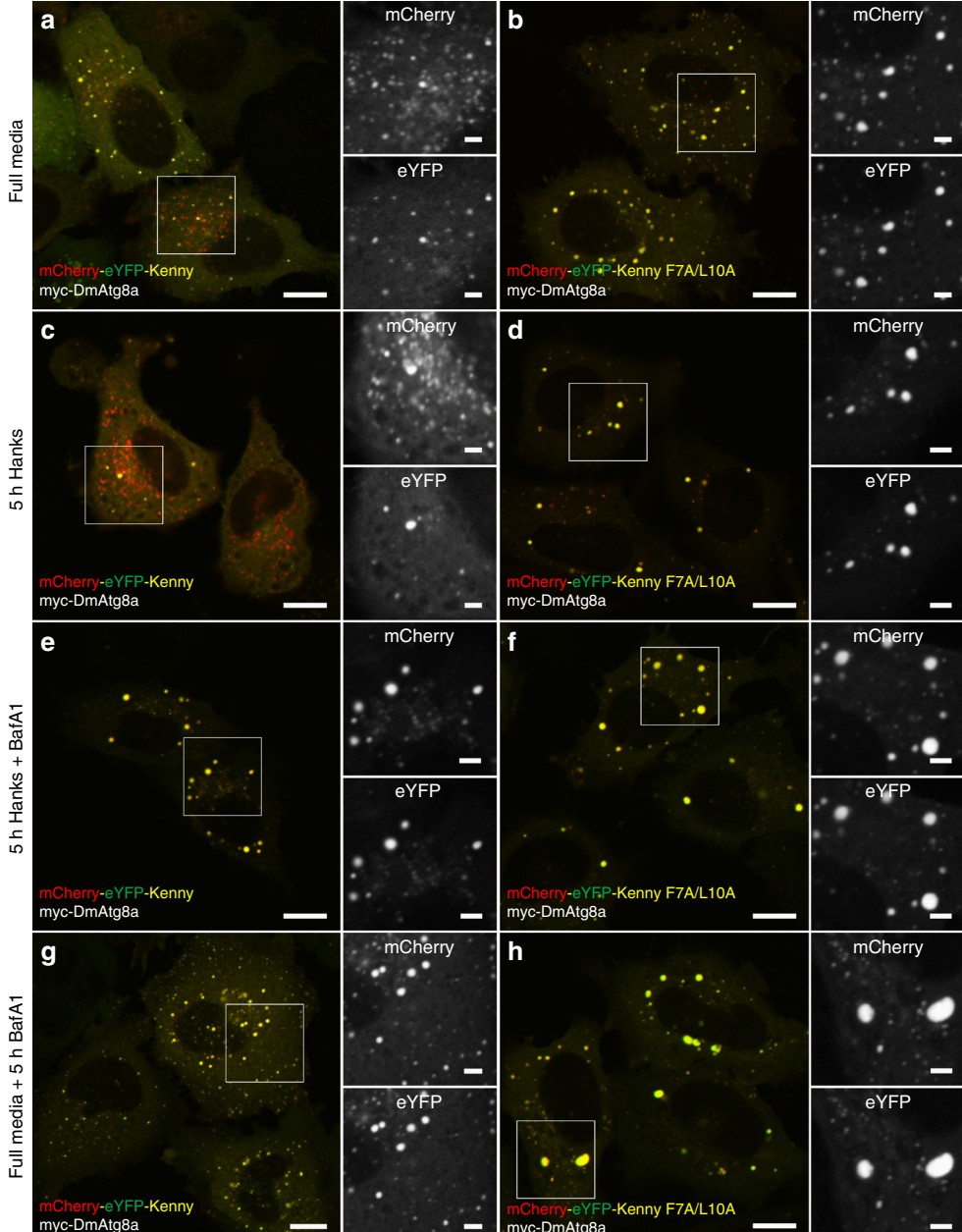

**Fig. 5** Autophagic degradation of Kenny in HeLa cells depends on a functional LIR motif and its interaction with co-expressed Atg8a. Confocal images of HeLa cells transiently expressing tandem tagged, mCherry-eYFP-Kenny^WT or F7A/L10A (red and green) and Myc-Atg8a. **a**, **b** Cells cultured in full media. **c**, **d** Cell starved for 5 h in Hanks media to activate autophagy. **e**, **h** To inhibit lysosomal degradation, the cells in full media **e**, **f** or Hanks media **g**, **h** were treated with Bafilomycin A1 (BafA1). Insets on the right of merged channels show the separate red and green channels. Scale bars are 10 and 2 μm (insets)

immune responses by the microbiota, and host types 3 and 4 cannot succeed because they bear the extra cost of maintaining pathways to autophagically degrade IKKγ by ubiquitin tagging.

As soon as we make additional assumptions about host or pathogen advantage during infection, it becomes possible to obtain a human-like scenario in which host type 3 dominates. If we make the assumption that the host is best able to regulate immunopathology during infection if IKKγ is regulated using ubiquitin tagging, and that this advantage is greatest if IKKγ is not being targeted for degradation by a LIR motif (Effect II), host type 3 will dominate (Fig. 9a, bottom row). Alternatively, if there exists a pathogen which encodes LIR, for which infection mortality is specifically reduced in hosts where IKKγ lacks LIR (Effect IIIb), theoretically this alone could overcome the cost of

maintaining pathways to degrade IKKγ by ubiquitin tagging and allow host type 3 to dominate (Fig. 9a, top right hand panel). There are therefore at least two biologically plausible mechanisms by which host-pathogen co-evolution could result in the loss of the LIR motif from IKKγ.

## Discussion

Multicellular organisms are exposed to microbes throughout their lives. These exposures can be transient or permanent, with microbes ranging from those considered commensal to pathogenic[38–41]. The initial host response to these microbes is mediated by the innate immune system. This highly conserved host defense has to be tightly controlled in order to not become

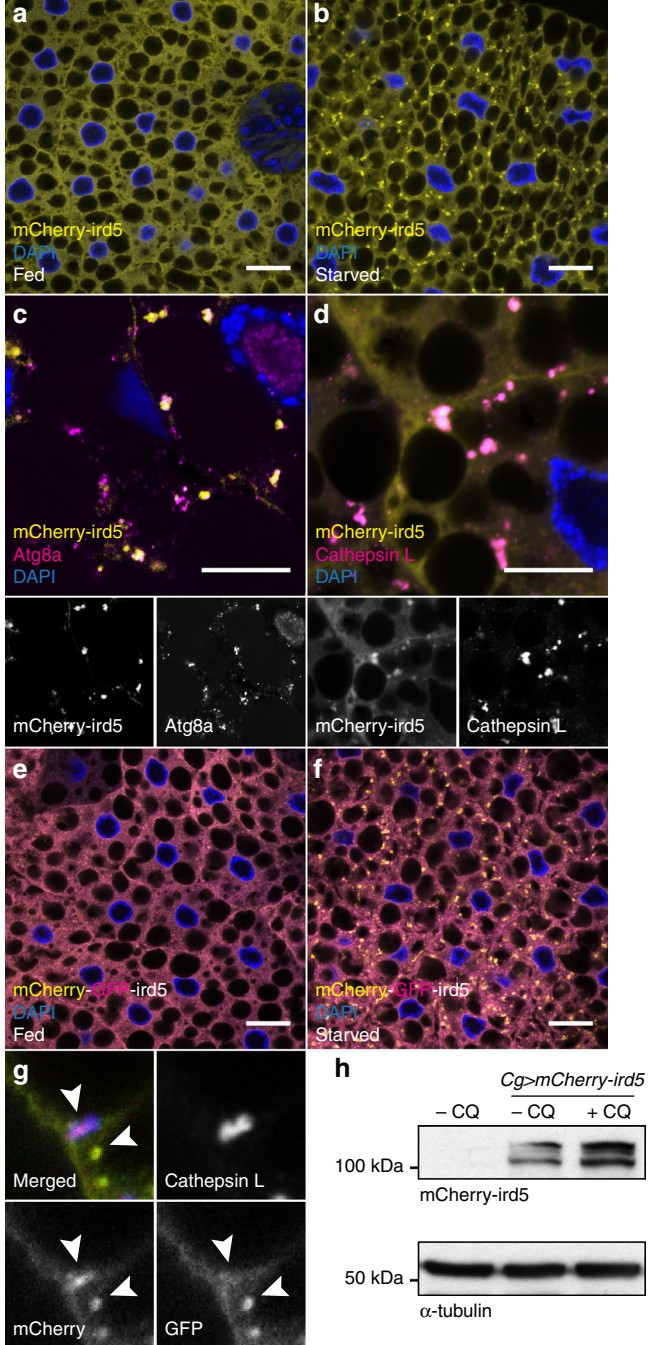

**Fig. 6** ird5/IKKβ is degraded by autophagy. **a**, **b** Confocal images of fat body cells from fed **a** and starved **b** larvae expressing mCherry-ird5 (yellow). **c**, **d** Co-localization between mCherry-ird5 (yellow) and Atg8a (magenta) **c** or Cathepsin-L (magenta) **d**. Insets show the separate channels for mCherry-ird5 (lower left), Atg8a (lower right), and Cathepsin-L (lower right). **e**, **f** Confocal images of fat body cells from fed **e** and starved **f** larvae expressing a tandem-tagged mCherry-GFP-ird5 (yellow and magenta). **g** High magnification of fat body cells from starved larvae expressing mCherry-GFP-ird5 (green and red, lower panels), stained for Cathepsin-L (upper right panel). Arrowheads show ird5 in autophagosomes (both red and green signals co-localize) and in autolysosomes (only the red signal is detectable due to GFP quenching). Scale bars are 20 μm **a**, **b**, **e**, **f** and 10 μm **c**, **d**. **h** Western blot analysis of lysates from larvae fed for 25 h with food supplemented with 2.5 mg/mL Chloroquine (+CQ) or vehicle (−CQ)

harmful through inflammation[38–41]. In this study we have described an essential role of selective autophagy in the termination of the innate immune response to commensal microbiota in *Drosophila*. This process is mediated by Kenny which acts as a selective autophagy receptor for the autophagic degradation of the IκB kinase (IKK) complex. Kenny is a component of the immune deficiency (IMD) pathway which is activated following the recognition of bacterial diaminopimelic acid peptidoglycan (DAP-PGN), leading to the assembly of a receptor proximal signaling complex[5,23,42,43]. Downstream signaling requires the cleavage of the nuclear factor-kappa B (NF-κB)-like immune transcription factor Relish, which translocates to the nucleus and induces the expression of antimicrobial peptide (AMP) genes[30–32,44]. Here we showed that selective autophagic degradation of the IKK complex prevents the systemic immune response that is mediated by Relish and the production of antimicrobial peptides. Autophagy mutants fail to terminate this response and exhibit systemic inflammation that contributed to their premature death. *Atg8a* and *Atg7* mutant flies have been reported to have a short lifespan due to a neurodegeneration phenotype[45,46]. Our results indicate that their short lifespan is also mediated by systemic inflammation that affects the function of the gut.

Selective autophagic degradation of Kenny is mediated by its LC3-interacting region (LIR) motif. We have shown the importance of the LIR motif for its degradation using four different technical approaches: (1) mutation of the LIR motif drastically reduces the co-immunoprecipitation between Kenny and Atg8a when expressed in S2R+ cells, (2) mutation of the Kenny LIR motif abrogates the direct interaction with Atg8a, as demonstrated by GST pull-down experiments. Similarly, Kenny loses its interaction with Atg8a lacking a functional LIR docking site (LDS). (3) Using tandem-tagged Kenny expressed in HeLa cells, we observed that wild-type Kenny is targeted to acidic compartments (autolysosomes) when the cells are under starved conditions, while its LIR mutated counterpart is not, and (4) the ability of GFP-Kenny to localize to autophagosomes and autolysosomes in vivo is compromised when its LIR motif is mutated. Although we were not able to show that autophagic degradation of a genomic Kenny LIR mutant was compromised, we have clearly shown that endogenous Kenny protein was robustly accumulated in *Atg8a* and *Atg7* mutant flies.

Furthermore, we showed that Kenny is accumulated in autophagy mutant flies in its phosphorylated form. Interestingly, Kenny does not appear to be phosphorylated when the proteasome is inhibited. This suggests that phosphorylation of Kenny is important for its function as selective autophagy receptor. Indeed it has been shown that phosphorylation of various autophagy receptors enhances their autophagic degradation[47–52].

We showed that *Atg8a* and *Atg7* mutant flies exhibit intestinal hyperplasia that is predominantly controlled by a systemic immune response produced by the fat body. This is in line with a previous study showing that systemic inflammation caused by the fat body leads to intestinal hyperplasia in old flies[53]. Despite the fact that we observed a very modest upregulation of *Diptericin* in isolated autophagy mutant guts we have observed nuclear translocation of Relish in the midgut of *Atg8a* mutant flies. A report by Erturk-Hasdemir et al.[33] proposed that the IKK complex is required both for the cleavage and phosphorylation of Relish, the latter being responsible for the efficient upregulation of AMP genes. There is evidence however that Relish cleavage alone may be enough to upregulate the expression of *Diptericin* only[54]. Regulation of the IMD pathway is tissue-specific with its various output molecules being favoured differently from one tissue to another[55]. Nuclear translocation of Relish in *Atg8a* mutants may lead to some level of *Diptericin* expression, however

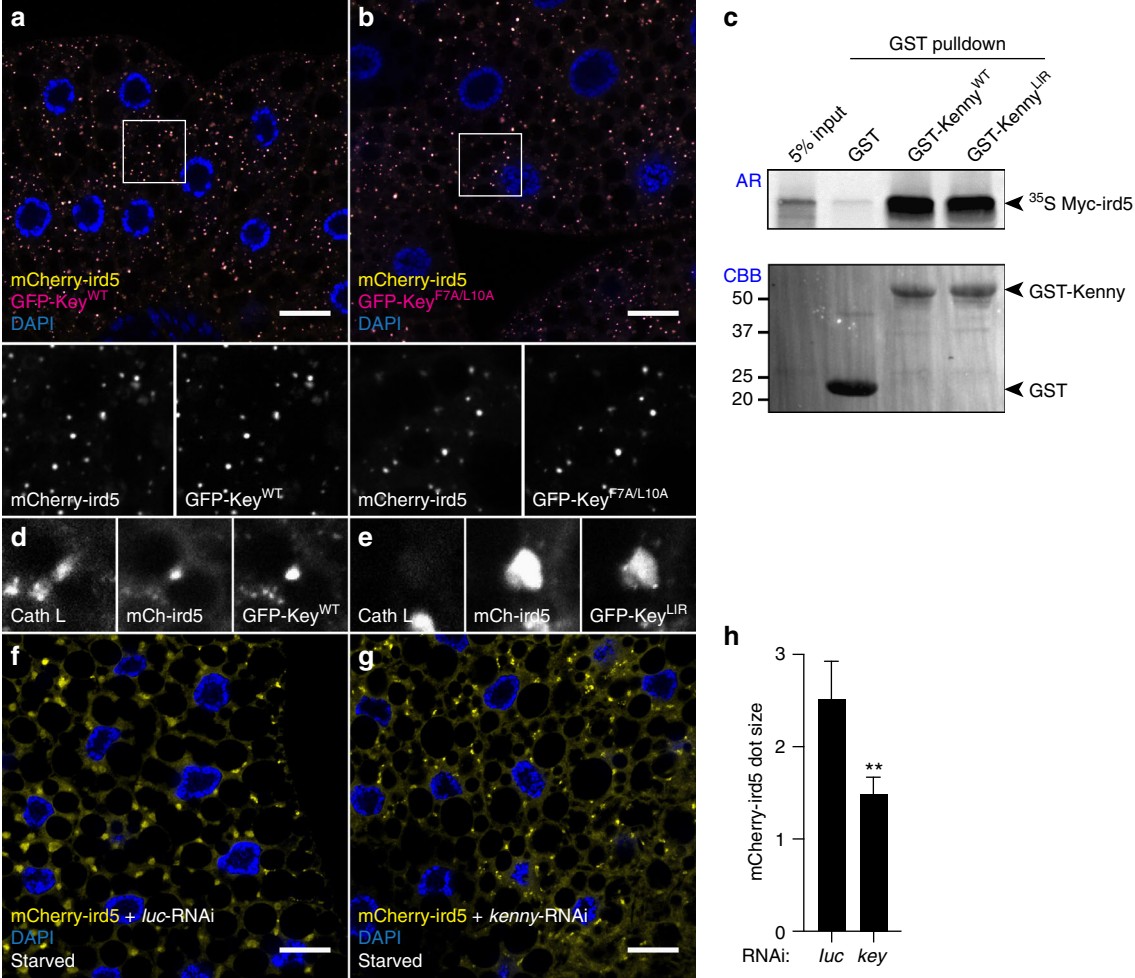

**Fig. 7** Localization of ird5 in lysosomes. **a**, **b** Confocal images of fat body cells from fed larvae co-expressing mCherry-ird5 (yellow) and GFP fusion proteins of Kenny[WT] **a** or Kenny[F7A/L10A] (magenta) **b**. Insets show the separate channels for mCherry-ird5 (lower left) and GFP-Kenny (lower right). **c** In vitro GST-pull-down assay between GST-tagged Kenny (lower panel, Coomassie Blue), and radiolabelled Myc-ird5 (upper panel) produced by coupled in vitro transcription and translation reaction in the presence of [35]S-methionine. **d**, **e** Magnification of confocal images of fat body cells from starved larvae co-expressing GFP-Kenny[WT] (**d**, GFP-Key[WT]) or GFP-Kenny[F7A/L10A] (**e**, GFP-Key[LIR]), stained for Cathepsin-L. **f**, **g** Confocal images of fat body cells from starved larvae co-expressing a control luciferase RNAi (**f**, luc-RNAi) or a RNAi against kenny (**g**, key RNAi) along with mCherry-ird5 (yellow). **h** Quantification of the size of the mCherry-ird5 puncta. Bar chart shows means ± s.d. Statistical significance was determined using two-tailed Student's t-test, **P < 0.01. Scale bars are 20 μm

in the absence of infection this may in fact favor the expression of the IMD pathway antagonist Pirk and prevent Relish phosphorylation, which would have an overall negative effect on the local production of AMPs[56].

It has been reported that Atg9 but not Atg12 was required for JNK-mediated intestinal hyperplasia in *Drosophila* under oxidative stress conditions (paraquat feeding)[57]. However, in this study knockdown of *Atg9* or *Atg12* was done tissue specifically in the midgut, whereas our results are from full body mutants and without paraquat feeding.

Interestingly we found that the mammalian homolog of Kenny, IKKγ/NEMO, does not interact with any of the mammalian Atg8-family members, so cannot be directly targeted for autophagy by such interactions. There is evidence that IKK degradation may be able to be triggered by proteins expressed by mammalian pathogens. The bacterial effector geldanamycin, which is an inhibitor of Hsp90, causes degradation of the IKK complex by autophagy[58]. Murine cytomegalovirus M45 protein interacts with NEMO, mediating its engulfment by the autophagosome and subsequent degradation in the lysosome[35]. Analysis of the murine cytomegalovirus M45 protein sequence with iLIR[20] shows a

predicted functional LIR motif. It has also been shown that p47, an essential factor for Golgi membrane fusion, associates with NEMO upon tumor necrosis factor (TNF) or interleukin-1 stimulation, and inhibits IKK activation. p47 binds to NEMO and mediates its lysosomal degradation[59]. Interestingly, Shp1, a yeast homolog of p47, binds to ATG8 which is the yeast homolog of LC3 and is required for the elongation of the nascent autophagosome[60]. Altogether, these results suggest that there was a switch during the evolution of the IKKγ protein in metazoans. *Drosophila* IKKγ/Kenny can be directly and selectively degraded by autophagy through its functional LIR motif. In contrast, the targeting of mammalian IKKγ for autophagic degradation seems to be mediated either by bacterial effectors (geldanamycin), viral proteins (M45) or through putative selective autophagy receptors (p47).

To investigate what kinds of interactions between hosts, pathogens, and the microbiota could drive the loss of a LIR motif from IKKγ, we developed a mathematical model in which the mechanisms by which host IKKγ could be selectively targeted for autophagic destruction varied. Hosts of different types competed in the presence of pathogens and a proxy for the microbiota, and

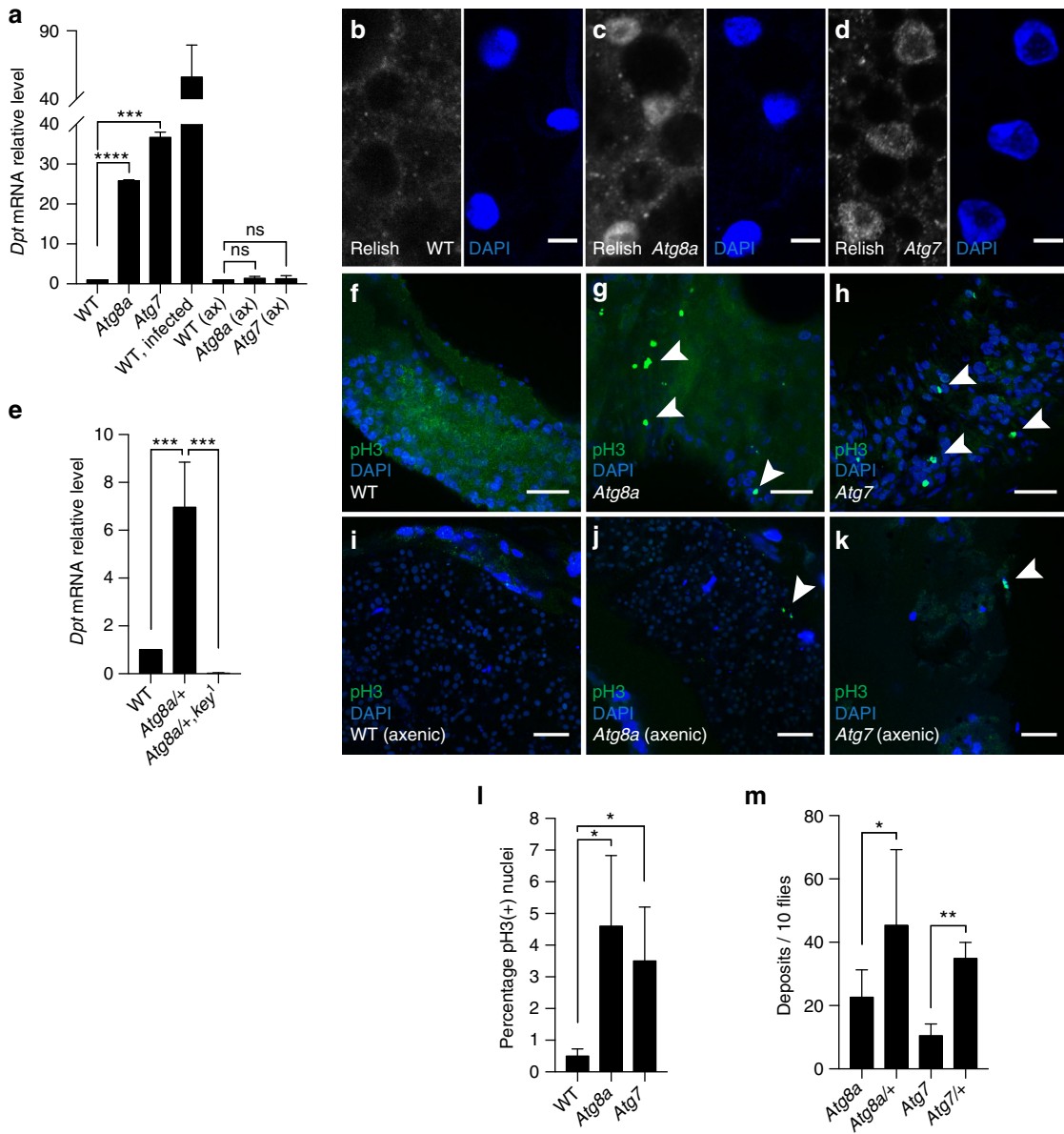

**Fig. 8** Deregulation of the IMD pathway and intestinal dysplasia in autophagy-deficient flies is induced by commensal bacteria. **a** Analysis of *Dpt* mRNA levels in flies reared in conventional or axenic conditions. **b–d** Confocal images of fat bodies from conventionally reared wild-type **b**, *Atg8a* **c** and *Atg7* **d** mutant adult flies stained for Relish (gray) and nuclei (blue). Scale bars are 5 μm. **e** Analysis of *Dpt* mRNA levels in conventionally reared Atg8a/Kenny double mutant flies. **f–k** Confocal images of posterior midguts stained for phospho-H3 (pH3, green) from wild-type **f**, **i**, *Atg8a* **g**, **j** and *Atg7* **h**, **k** mutant flies reared in conventional **f–h** or axenic conditions **i–k**. Arrowheads show pH3-positive cells. Scale bars are 20 μm. **l** Quantification of the percentage of pH3-positive cells per picture. **m** Quantification of the number of deposits per 10 flies. Bar charts show means ± s.d. Statistical significance was determined using one-way ANOVA, *$P < 0.05$, **$P < 0.01$, ***$P < 0.001$, ****$P < 0.0001$

we investigated which selective processes caused one or another host type to dominate the population. Our model highlighted two potential mechanisms by which pathogen selection could drive the loss of a LIR motif from IKKγ. The first assumes that hosts which regulate IKKγ using ubiquitin tagging alone have the lowest infection mortality, because they are best able to regulate immunopathology during infection. The second mechanism involves a specific reduction in infection mortality where pathogens encode their own LIR motifs, that only applies in hosts which lack a LIR motif on IKKγ—in other words, pathogen manipulation of the host using LIR motifs, which happens to benefit both host and pathogen by reducing infection mortality.

For simplicity, our model pitted the four different host types against one another, starting with each host type at an equal

frequency. This showed us which host type was best suited to a particular set of selective pressures, but meant that each simulation started with an evolutionarily unrealistic situation in which equal numbers of hosts with or without the ability to regulate IKKγ using ubiquitin tagging were present. In reality, evolving the ability to regulate IKKγ using ubiquitin tagging must have required a series of steps, including gaining appropriate selective autophagy receptors, before this mechanism to regulate IKKγ could have emerged. It is entirely plausible that pathogen selection was one of the selective pressures which made the evolution of this machinery favorable—but our model does not explicitly capture how the machinery evolved. Considering the severely negative consequences of a complete inability to regulate IKKγ (as captured by our experimental results), we can speculate that some

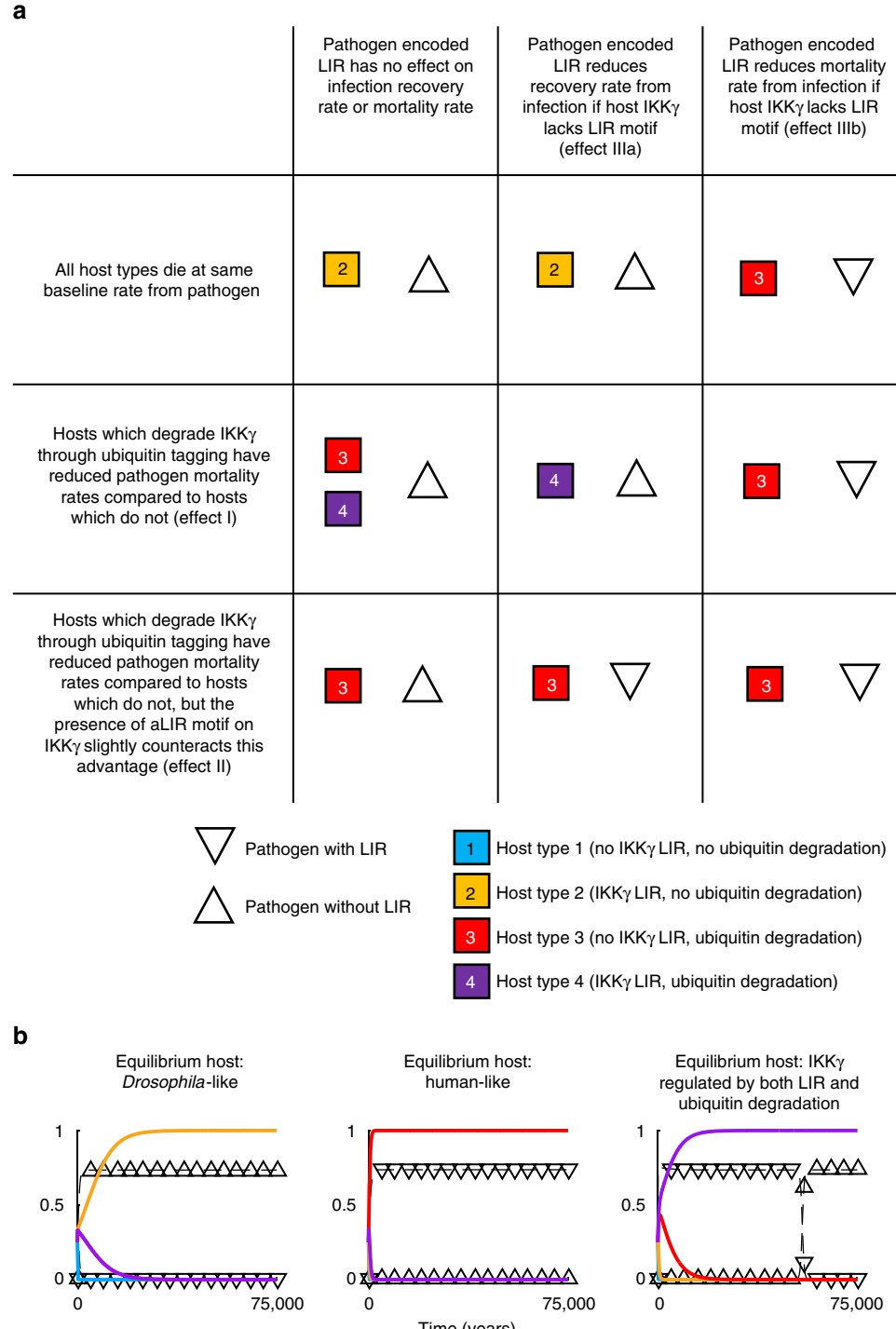

**Fig. 9** A mathematical model of LIR motif functionality in co-evolving hosts and pathogens. **a** The equilibrium outcomes when different assumptions about infection recovery rate and mortality (Effects I–IIIb described in the main text) are applied separately or in combination. The host type(s) present at equilibrium are indicated by the different colored and numbered squares and the pathogen types by triangles. An upright triangle indicates the pathogen circulating at equilibrium can express a LIR motif, a downturned triangle indicates the pathogen does not express a LIR motif. For a description of the model and all parameter values, see Methods section and Supplementary Table 5. **b** Time series showing the dynamics leading up to equilibrium under three different sets of conditions: applying none of the effects in the main text, resulting in a *Drosophila*-like scenario; applying Effects I and IIIb leading to a human-like host and a LIR expressing pathogen at equilibrium; and applying Effects I and IIIa, leading to an equilibrium population where host IKKγ is regulated by both LIR and ubiquitin degradation. The different colored lines show the proportion of the host population belonging to a particular host type and the triangular markers indicate the proportion of the population infected with a pathogen that can express LIR (downturned triangle) or a pathogen that cannot express LIR (upturned triangle)

alternative means to regulate IKKγ is a prerequisite before the LIR motif can be lost from IKKγ, and thus that the ability to regulate IKKγ using ubiquitin tagging in mammals may have evolved before any loss of the LIR motif.

This point is particularly pertinent when considering the second of the two evolutionary scenarios we found that could drive the loss of LIR from IKKγ. In that scenario there is a specific reduction in infection mortality where pathogens encoding their own LIR motifs infect hosts which lack a LIR motif on IKKγ. We considered this an important result to include, by way of showing that pathogens with very particular properties could drive the host to lose the LIR motif from IKKγ; however, reaching that equilibrium point requires that there already exist some hosts in the population for which there is not an excessive cost to losing the LIR motif (i.e., hosts with some other mechanism to regulate IKKγ, such as ubiquitin tagging). Only after such hosts lose their LIR motifs can there be a niche for the pathogen with the necessary properties to invade the population in the first place, and drive the evolutionary switch whereby the entire host population ends up losing the LIR motif from IKKγ.

The first of the evolutionary scenarios we proposed has less restrictive conditions. If hosts which regulate IKKγ using ubiquitin tagging alone have the lowest infection mortality, because they are best able to regulate immunopathology during infection, then selection from pathogens with or without LIR motifs can drive a population to an equilibrium where the host loses the LIR motif from IKKγ.

In conclusion, we have shown that autophagy plays a critical role in the termination of innate immune signaling in response to commensal microbiota in *Drosophila* by degrading Kenny and ird5. We cannot exclude that other components of the IMD pathway are also degraded by autophagy through interaction with Atg8a or other autophagy-related proteins. Further studies are awaited to clarify these questions. Our study highlights the physiological importance of selective autophagy in the innate immune response of metazoans, and demonstrates the plasticity of its participating regulators. We propose that a major switch in the regulation of IKKγ occurred during metazoan evolution, and have shown how this switch could have been driven by host-pathogen coevolution.

## Methods

**Fly husbandry and generation of transgenic lines**. Flies used in experiments were kept at 25 °C and 70% humidity raised on cornmeal based feed.

The following fly stocks were obtained from the Bloomington *Drosophila* stock center: *w1118* (#3605), *UAS-Atg5-RNAi* (#27551), and *Cg-GAL4* (#7011). *UAS-Atg8a-RNAi* (#109654) and *UAS-key* RNAi (#7723) were obtained from the Vienna *Drosophila* Ressource Center. The following mutant lines have been used: *yw hs-FLP Atg8aKG07569*, *atg7Δ77*, and *atg7Δ14/CyO-GFP*[46] (gift from G. Juhasz), *key123* (gift from D. Ferrandon), *yw hs-flp;FRT82 atg13Δ8161* and *yw hs-flp;FRT80 atg1Δ3D*[62]. For the mutant mosaic analysis (FLP/FTR and MARCM mitotic recombination), the following lines were used: *yw hs-flp;FRT82 Ubi::GFP* and *yw hs-flp tub-Gal4 UAS-GFP; FRT80B tub-Gal80 hsc82 y+* (gift from Y. Fan). The clonal analysis using the FLPout system have been performed with the following lines: *yw hs-flp Ac > CD2 > GAL4 UAS-GFP* (gift from G. Juhasz) and *yw hs-flp; UAS-mCherry-Atg8a;Ac > CD2 > GAL4*. The transgenic lines *UAS-GFP-KennyWT*, *UAS-GFP-KennyF7A/L10A* and *UAS-GFP-NEMO* have been generated by cloning the cDNA of Kenny or NEMO respectively into the pPGW plasmid (DGRC). The transgenic line *UAS-mCherry-ird5* has been generated by cloning the cDNA of ird5 into the pP(mCherry)W plasmid modified from the pPGW by replacing the sequence coding for the GFP by mCherry. The transgenic line *UAS-mCherry-GFP-ird5* has been generated by cloning the cDNA of ird5 into the pP(mCherry)GW plasmid modified from the pPGW backbone by adding the sequence coding for mCherry in 5′ of the GFP. Transgenic flies were generated by P-element-mediated transformation (BestGene Inc). The genotype of the flies used for the experiments displayed in Figs 2–8 are listed in Supplementary Table 1.

**Clonal analysis (FLP/FRT mosaic, MARCM, FLPout techniques)**. For the generation of mutant clones in the larval fat body using the FLP/FRT (flippase mediated site-specific recombination) and MARCM (Mosaic analysis with a repressible cell marker) mitotic recombination techniques, embryos 0–10 h after

egg laying were subjected to heat shock for 1 h at 37 °C. For the FLPout UAS/GAL4 method, the GAL4 expressing cells were generated spontaneously without heat shock[25]. L3 larvae at 4 days after egg laying were dissected.

**Generation of axenic flies**. Embryos were collected on apple juice plates for 10 h and dechorionated in 2.7% bleach for 2 min, rinsed in 70% ethanol twice and then in distilled water twice. Embryos were then transferred onto food containing a cocktail of antibiotics. Larvae and adults were reared on autoclaved cornmeal food, enriched with either 0.25 mg/mL ampicillin, tetracycline, streptomycin, and 1 mg/mL kanamycin. Axenic status of the flies was confirmed by CFU on LB agar plates, with one full fly body smear per plate and 16 S PCR (forward primer: 5′-CAGGCCTAACACATGCAAGTC-3′; reverse primer: 5′-ACGGGCGGTGTG-TACAAG-3′). At least 2 flies per stock tube were used to attest the axenic status of the flies prior to experiment.

**Bortezomib feeding**. Adult flies were transferred within 24 h from hatching onto Nutri-Fly Instant *Drosophila* Medium (Genesee Scientific, 66–117) prepared in water supplemented with 5–20 μM Bortezomib or DMSO (max 0.2% final). Flies were flipped into freshly made food every other day for 5–6 days.

**Protein extraction, immunoprecipitation, and western blotting**. Protein content was extracted from the full fly body in TNT lysis buffer (40 mM Tris pH 7.4, 150 mM NaCl, 1% Triton X-100, 2 mM EDTA, 1 mM Na3VO4, 5 mM Na4P2O7, 50 mM NaF + EDTA-free proteases inhibitors cocktail) using a motorized mortar and pestle. Co-immunoprecipitations were performed on lysates from flies expressing GFP alone or GFP-Kenny along with mCherry-ird5. After a 30 min pre-clear of the lysates (1 mg total proteins) with sepharose-coupled G beads (Sigma), the co-immunoprecipitation was performed for 2 h at 4 °C using an anti-GFP antibody (Abcam, Ab290) and fresh sepharose-coupled G-beads. Four consecutive washes with the lysis buffer were performed before suspension of the beads in 60 μL 2X Laemmli loading buffer. For ubiquitination assay, lysates were boiled for 5 min at 95 °C in TNT lysis buffer supplemented with 2% SDS final. Denatured lysates were then diluted in lysis buffer without SDS to reach a final SDS concentration that does not exceed 0.2% prior to immunoprecipitation overnight at 4 °C. In addition to the usual proteases inhibitor cocktail, the lysis buffer was supplemented with the deubiquitinating enzymes specific inhibitor PR-619. All protein samples (whole fly lysates and co-immunoprecipitation eluates) were boiled for 5 min at 95 °C. Quantity of 10–40 μg of proteins (whole fly lysates) or 20 μL (co-immunoprecipitation eluates) were loaded on acrylamide gels and were transferred onto either nitrocellulose or PVDF membranes (cold wet transfer in 10% ethanol for 1 h at 100 V, or 2 h at 30 V for the detection of ubiquitin chains).

Membranes were blocked in 5% BSA or non-fat milk in TBST (0.1% Tween-20 in TBS) for 1 h. Primary antibodies diluted in TBST were incubated overnight at 4 °C or for 3 h at room temperature with gentle agitation. HRP-coupled secondary antibodies binding was done at room temperature (RT) for 45 min in 1% BSA or non-fat milk dissolved in TBST and ECL mix incubation for 2 min. All washes were performed for 10 min in TBST at RT.

The following primary antibodies were used: anti-Kenny[23] (gift from Dr N. Silverman, 1:5000), anti-GFP (Santa Cruz sc-9996, 1:1000), anti-mCherry (Novus NBP1–96752, 1:2000), anti-alpha tubulin (Sigma-Aldrich T5168, 1:40,000), anti-ubiquitinated proteins (clone FK2, Enzo Life Sciences BML-PW8810, 1:200), anti-ubiquitin K48 specific (clone Apu2 Sigma-Aldrich 05–1307, 1:1000). HRP-coupled secondary antibodies were from ThermoScientific (anti-mouse HRP #31450; anti-rabbit HRP #31460). Following co-immunoprecipitation, Veriblot HRP-coupled IP secondary antibody was used (Abcam ab131366, 1:5000).

The uncropped scan of the original films used to assemble the main figures are available in Supplementary Fig. 17.

**Dephosphorylation assay**. Lysates from *Atg8aKG* and *Atg7Δ14/Δ77* adult flies were prepared in modified TNT buffer (40 mM Tris pH 7.4, 150 mM NaCl, 1% Triton X-100 + EDTA-free proteases inhibitors cocktail). Dephosphorylation of 50 μg of total protein was performed in 50 μL reaction volume (top up with ddH2O supplemented with EDTA-free proteases inhibitors cocktail) in the presence of either 10 U/μg proteins of CIP (Calf intestinal alkaline phosphatase; NEB, M0290S) or 8 U/μg proteins of λPP (lambda protein phosphatase; NEB, P0753S) supplemented with their respective reaction buffers according to manufacturers' protocol. As negative control, phosphatase inhibitor cocktail 2 (Sigma, P5726) at 1:50 was added to some reactions. All the samples were then incubated at 37 °C for 15 min before dilution in Laemmli buffer and heating for 10 min at 80 °C.

**Immunohistochemistry**. Fly tissues were dissected in PBS and fixed for 30 min in 4% formaldehyde. Blocking and antibody incubations were performed in PBT (0.3% BSA, 0.3% Triton X-100 in PBS). Primary and secondary antibodies were incubated overnight at 4 °C in PBT. The following primary antibodies were used: anti-Cathepsin-L (Abcam ab58991, 1:400), anti-GABARAP (Cell Signaling Technology #13733, 1:400), anti-pH3 (Millipore #06-570, 1:1000), anti-Kenny[23] (gift from Dr N. Silverman, 1:600), anti-Relish (#Abin1111036, RayBiotech 130-10080, 1:300). Specificity of the anti-Relish antibody was tested using

Relish E20 mutants[63] (Supplementary Fig. 18). Washes were performed in PBW (0.1% Tween-20 in PBS). All images were acquired using Carl Zeiss LSM710 or LSM880 confocal microscopes, using a ×63 Apochromat objective. Images were post-processed in Fiji for co-localization studies.

**Real-Time qPCR.** RNA extraction was performed with a Life Technologies Ambion PureLink™ RNA Mini kit according to the manufacturer protocol. Ten full adult flies or 50 isolated guts were used per extract. Males were preferentially used, but due to the lack of male mutants for the double mutant flies $Atg8a;key^1$, only heterozygote females for $Atg8a$ were used.

Subsequent steps were performed using 1 µg of total RNA. ThermoScientific DNase I was used in order to digest genomic DNA. The ThermoScientific RevertAid Kit was subsequently used to synthesize cDNA. RT-qPCR was performed using the Promega GoTaq qPCR Master Mix (ref. A6002). Primer sequences are available in Supplementary Table 2.

**Lifespan measurement.** In order to measure lifespan, males only were collected within 24 h from hatching and cohorts of 20–25 flies were maintained on standard or autoclaved/antibiotics-supplemented *Drosophila* food at 25 °C in a humidified incubator. Flies were transferred into new tubes every 2–3 days. Dead events were recorded daily. Axenic status of the flies was checked once a week. The statistical analysis has been done with Prism7 software (GraphPad), using a Gehan-Breslow-Wilcoxon test.

**Plasmid constructs.** Plasmids used in this study are listed in Supplementary Table 3. cDNAs were made using Transcriptor Universal cDNA Master (Roche Applied Science, 05 893 151 001). PCR products were amplified from cDNA using Phusion high fidelity DNA polymerase with primers containing the Gateway recombination site or restriction enzyme sites for Gateway entry vector and cloned into pDONR221 or pENTR using Gateway recombination cloning. Plasmids were made by conventional restriction enzyme-based cloning and/or by use of the Gateway recombination system (Invitrogen). Point-mutants were generated using the QuikChange site-directed mutagenesis (Stratagene, 200523). Oligonucleotides for mutagenesis, PCR and DNA sequencing were from Invitrogen or Sigma. Plasmid constructs were verified by conventional restriction enzyme digestion and/ or by DNA sequencing with BigDye (Applied Biosystems, 4337455).

**Cell culture and immunoprecipitation in *Drosophila* S2R cells.** S2R+ cells (DGRC, Indiana University, Bloomington, IN) were cultured at 25 °C in Schneider's Drosophila medium (Gibco/Invitrogen, 21720-024) supplemented with 10% heat-inactivated fetal bovine serum and 1% streptomycin-penicillin (Sigma, P4333). Subconfluent cells were transfected with plasmids using TransIT-2020 (Mirus, MIR5400) following the supplier's instructions and analyzed 48 or 72 h after transfection. For immunoprecipitation, S2R+ cells were washed 72 h after transfection with ice-cold PBS prior to lysis in lysis buffer (50 mM Tris-HCl, pH 7.5, 150 mM NaCl, 1 mM EDTA, 1% Nonidet P-40 (v/v), 0.25% Triton X-100(v/v)) supplemented with phosphatase inhibitor mixture set II (Calbiochem) and Complete Mini, EDTA-free protease inhibitor mixture (Roche Applied Science, 11836170001). Lysates were incubated with anti-FLAG M2-agarose beads (Sigma, A2220) at 4 °C overnight. Beads were washed five times with 500 µl lysis buffer. Proteins were eluted using 3XFLAG fusion peptide (Sigma, F4799), boiled in SDS-PAGE loading buffer, and subjected to western blotting. The following antibodies were used: mouse anti-FLAG antibody (Stratagene, 200471), rabbit polyclonal GFP antibody (Abcam, Ab290).

**GST Pull-down assays.** GST-fusion proteins were expressed in *Escherichia coli* BL21(DE3) and/or SoluBL21 (Amsbio) and GST-fusion proteins were purified on glutathione-Sepharose 4 Fast Flow beads (Amersham Biosciences). GST pull-down assays were performed using in vitro translated $^{35}$S-methionine-labelled proteins. L-[$^{35}$S]-methionine was obtained from PerkinElmer Life Sciences. A volume of 10 µL of the in vitro translation reaction products (0.5 µg of plasmid in a 25 µL reaction volume) were incubated with 1–10 µg of GST-recombinant protein in 200 µL of NETN buffer (50 mM Tris, pH 8.0, 100 mM NaCl, 6 mM EDTA, 6 mM EGTA, 0.5% Nonidet P-40, 1 mM dithiothreitol supplemented with Complete Mini EDTA-free protease inhibitor cocktail (Roche Applied Science)) for 1 h at 4 °C, washed six times with 1 ml of NETN buffer, boiled with 2X SDS gel loading buffer, and subjected to SDS-PAGE. Gels were stained with Coomassie Blue and vacuum-dried. $^{35}$S-Labeled proteins were detected on a Fujifilm bio-imaging analyzer BAS-5000 (Fuji).

**Double-tag analyses of DmKenny in HeLa cells.** HeLa (ATCC CCL2) cells were grown in Dulbecco's modified Eagle's medium (DMEM). Cultured HeLa cells were maintained at 37 °C with 95% air and 5% $CO_2$ in a humidified atmosphere. HeLa cells were cultured in 8-well chambered coverslides (Nunc) and transiently transfected with 100 ng of pDest-mCherry-eYFP-Kenny WT or F7A/L10A constructs and 50 ng of pDestMyc-DmAtg8a using TransIT-LT1. Cells were fixed in 4% paraformaldehyde 24 h after transfection. Images were obtained using a Zeiss

LSM780 microscope. Quantifications of red only punctae per cell were done using Volocity 6.3 software (PerkinElmer).

**Statistics.** Statistical analyses were done with Prism6/7 software (GraphPad). For the comparison of two groups, two-tailed *t*-test was used. To compare three or more groups, one-way ANOVA with Dunnett's test correction was used. Statistical significance of fly survival was calculated using a log-rank Mantel-Cox test. The statistical description for the results shown in Figs 2–8 are listed in Supplementary Table 4.

**Mathematical model.** We combined epidemiological processes and host population genetics in a single dynamical model to allow different host genotypes to compete in the presence of various infectious agents. Epidemiological models incorporating an evolving host have previously been used to address questions of host-pathogen coevolution[64]. Our complete model is described by Eqs. 1–6, given below. Hosts could be infected with three different types of microorganism: a proxy for the microbiota, (M); a pathogen which does not express the LIR motif (P), and an otherwise identical pathogen variant which does express the LIR motif (Q). We assumed that coinfection between M and either P or Q was possible but that coinfection between P and Q was not possible. Upon recovery from infection with P or Q the host is once again susceptible to reinfection with P or Q. We allowed there to exist four different host genotypes (1–4). Host genotypes 1 and 2 are assumed to lack the ability to target IKKγ for autophagic degradation using a ubiquitin tag; host genotypes 3 and 4 are assumed to possess this ability. IKKγ carries a LIR motif in host genotypes 2 and 4 but not in host genotypes 1 and 3.

Infection with M is lifelong. Infection with P or Q has duration $1/\sigma_{x,i}$, where $\sigma_{x,i}$ represents the recovery rate from infection with pathogen x for host genotype i. All host genotypes experience a background death rate μ, making the average host lifetime in the absence of infection $= 1/\mu$. The cost of maintaining the necessary molecular machinery to target IKKγ for autophagic degradation using ubiquitin tagging is represented by an extra mortality rate $\gamma_i$, which depends on host genotype i (where genotypes lacking this molecular machinery have $\gamma_i = 0$). Excess mortality due to stimulation of the host immune response by the microbiota is captured by the death rate $\theta_i$, and excess mortality due to infection of host type i by pathogen x is captured by rate $\psi_{xi}$.

The rate of change of numbers of susceptible hosts of genotype i ($S_i$) is given by Eq. 1:

$$\frac{dS_i}{dt} = b_i + \sigma_{P,i}P_i + \sigma_{Q,i}Q_i - (\lambda_P + \lambda_Q + \lambda_M + \mu + \gamma_i)S_i \qquad (1)$$

The rate of change of numbers of hosts of genotype i infected with P only ($P_i$) is given by Eq. 2:

$$\frac{dP_i}{dt} = \lambda_P S_i - (\lambda_M + \sigma_{P,i} + \psi_{Pi} + \gamma_i + \mu)P_i \qquad (2)$$

The rate of change of numbers of hosts of genotype i infected with Q only ($Q_i$) is given by Eq. 3:

$$\frac{dQ_i}{dt} = \lambda_Q S_i - (\lambda_M + \sigma_{Q,i} + \psi_{Qi} + \gamma_i + \mu)Q_i \qquad (3)$$

The rate of change of numbers of hosts of genotype i infected with M only ($M_i$) is given by Eq. 4:

$$\frac{dM_i}{dt} = \lambda_M S_i + \sigma_{P,i}K_i + \sigma_{Q,i}L_i - (\lambda_P + \lambda_Q + \theta_i + \gamma_i + \mu)M_i \qquad (4)$$

The rate of change of numbers of hosts of genotype i infected with M and P ($K_i$) is given by Eq. 5:

$$\frac{dK_i}{dt} = \lambda_P M_i + \lambda_M P_i - (\sigma_{P,i} + \psi_{Pi} + \theta_i + \gamma_i + \mu)K_i \qquad (5)$$

The rate of change of numbers of hosts of genotype i infected with M and Q ($L_i$) is given by Eq. 6:

$$\frac{dL_i}{dt} = \lambda_Q M_i + \lambda_M Q_i - (\sigma_{Q,i} + \psi_{Qi} + \theta_i + \gamma_i + \mu)L_i \qquad (6)$$

The force of infection with each of the three pathogens ($\lambda_M$, $\lambda_P$, and $\lambda_Q$) are given by Eqs. 7–9, where βx is the transmission parameter for microorganism x, and N is the total size of the host population:

$$\lambda_M = \frac{\beta_M \sum_{i=1}^{i=4}(M_i + K_i + L_i)}{N} \qquad (7)$$

$$\lambda_P = \frac{\beta_P \sum_{i=1}^{i=4}(P_i + K_i)}{N} \qquad (8)$$

$$\lambda_Q = \frac{\beta_Q \sum_{i=1}^{i=4}(Q_i + L_i)}{N} \qquad (9)$$

The birth rate for host genotype i, $b_i$ is given by Eq. 10. Calculating the birth rate in this way ensures that the total population size remains constant—a simplifying assumption which effectively means the host population is regulated by

an external carrying capacity.

$$b_i = \frac{S_i + P_i + Q_i + M_i + K_i + L_i}{N}$$

$$\left( \mu N + \sum_{i=1}^{i=4} \left( \psi_{Pi}(P_i + K_i) + \psi_{Qi}(Q_i + L_i) + \theta_i(M_i + K_i + L_i) \right. \right.$$

$$\left. \left. + \gamma_i(S_i + P_i + Q_i + M_i + K_i + L_i) \right) \right) \tag{10}$$

When choosing the values of parameters, our purpose was not to simulate a specific host and pathogen, but rather to investigate the types of selective pressure that could potentially drive an evolutionary switch in LIR motif functionality. The absolute values of any of our parameters are therefore less important than their relative values.

Starting conditions for each simulation were always: 500 hosts of type 1, of which 494 were uninfected, 1 was infected with pathogen $P$, 2 with pathogen $Q$ and 3 with microbiota proxy $M$; 501 uninfected hosts of type 2; 502 uninfected hosts of type 3 and 503 uninfected hosts of type 4. We used the solver ode15s in Matlab R2015b to solve the system numerically and determine its behavior after 75,000 years.

In all simulations shown, the background death rate of the host ($\mu$) was equal to 0.083 per month. The transmission parameter for the proxy for the microbiota ($\beta_M$) was set at 0.5. The transmission parameter for pathogen variant P ($\beta_P$) was set at 3.01 and the transmission parameter for pathogen variant Q ($\beta_Q$) was set at 3, meaning that, all other things being equal, producing a LIR motif is costly to the pathogen and makes it slightly less likely to be transmitted. The cost of maintaining pathways so that IKKγ could be targeted for autophagic degradation by ubiquitin tagging was set at the arbitrary value of 0.0002 per months. This meant that $\gamma_1$ and $\gamma_2 = 0$, while $\gamma_3$ and $\gamma_4 = 0.0002$. The cost of an uncontrolled innate immune response to the microbiota was set at 0.1 per months, thus $\theta_1 = 0.1$ and $\theta_2$, $\theta_3$, and $\theta_4 = 0$.

Effects I, II, IIIa, and IIIb described in the main text were applied in the model by choosing specific values of parameters $\sigma_{x,i}$ and $\psi_{xi}$. If no Effect is being applied, all values of $\sigma_{x,i} = 0.7$ and all values of $\psi_{x,i} = 0.015$. The application of the different effects can be considered in terms of changes to these baseline values, summarized in the following paragraphs. All values of $\sigma_{x,i}$ and $\psi_{x,i}$ used in the simulations shown are also provided fully in Supplementary Table 5.

Effect I reduces the mortality rates of hosts which regulate IKKγ using ubiquitin tagging by a fixed value. For the results shown here, this reduction had a magnitude of 0.005.

Effect II is similar to Effect I, but hosts regulating IKKγ using ubiquitin tagging alone and hosts regulating IKKγ using both ubiquitin tagging and a LIR motif have their infection mortality rates reduced by different fixed values. For the results shown here hosts regulating IKKγ using ubiquitin tagging alone had an infection mortality rate reduction of 0.005 and hosts regulating IKKγ using both ubiquitin tagging and a LIR motif had an infection mortality rate reduction of 0.004.

Effect IIIa involves letting a pathogen expressed LIR motif slow the recovery rate from infection in hosts that do not have a LIR motif on IKKγ. For the results shown here, these specific recovery rates ($\sigma_{Q,1}$ and $\sigma_{Q,3}$) were given values of 0.6.

Effect IIIb involves reducing infection mortality by a fixed value (for the results shown here this reduction had a magnitude of 0.005), in the specific instance where the pathogen expresses a LIR motif and the host does not possess a LIR motif on IKK gamma. For the results shown here, this reduction in infection mortality rate is treated as independent to those of Effects I and II, and is added to them when the different effects are combined together.

**Data availability**. The authors declare that all data supporting the findings of this study are available within the article and its Supplementary Information files.

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

## Acknowledgements

We thank Dr N. Silverman for Kenny antibody, Drs D. Ferrandon, G. Juhasz and Y. Fan for providing fly stocks, and M. Ward for fly food preparation. The Bloomington *Drosophila* Stock Center and Vienna *Drosophila* Resource Center contributed to this work by providing mutant and transgenic fly strains. We acknowledge Bestgene Inc for the injection and selection of transgenic flies. The *Drosophila* Genomics Resource Center (supported by NIH grant 2P40OD010949) for providing the pPGW plasmid. We thank Gabor Juhasz for communicating results prior publication. This work was supported by BBSRC grants BB/L006324/1 and BB/P007856/1 awarded to I.P.N. R.T is supported by the Myrtle Pridgeon scholarship. T.J. was supported by grants from the Research Council of Norway (249884 and 196898) and from the Norwegian Cancer Society (71043-PR-2006-0320).

## Author contributions

I.P.N. conceived the project. R.T. and A.-C.J performed the in vivo experiments and statistical analysis. A.J. and K.B.L. performed the in vitro experiments. B.S.P. developed the mathematical model. I.P.N, T.J., R.T., A.J., and A.-C.J. designed the experiments. I.P. N., A.-C.J., R.T., and B.S.P. composed the manuscript. All the authors reviewed the manuscript and discussed the work.

## Additional information

**Competing interests:** The authors declare no competing financial interest.

