## [Peer Review File · Nature Communications]

Reviewers' comments:

Reviewer #1 (Drosophila IKK, innate)(Remarks to the Author):

Tusco et al. propose an interesting model, where they argue that in flies, Kenny (the fly IKKb/NEMO homolog) is selectively degraded by autophagy and this phenomenon is facilitated by the presence of AIM (Atg8 –Interacting-Motif) domain in Kenny. Furthermore, in autophagy mutants, up regulation of IMD pathway is observed and this results in the hyperplasia of gut, which ultimately leads to decreased fly survival.

Major Concerns

Although their findings are fascinating, there are some weaknesses and poorly defined experiments that detract from the conclusions. For example, they claim that up regulation of the IMD pathway in the gut causes hyperplasia, but do not show that IMD signaling is increased specifically in this tissue. Moreover, the experimental data fails to demonstrate the IMD signaling is causative to the cut hyperplasia in the autophagy mutants. In another claim, they argue that autophagy selectively degrades Kenny happens, but didn't determine if other IMD pathway components are also degraded. Just how selective this degradation is, or what are the important targets in terms of regulating IMD signaling are not clear; for example Figure 6C appears to show elevated Relish in autophagy mutants. In particular, it seems far-fetched that Kenny persistence alone, due to autophagy inhibition, would drive AMP expression, as high levels of Kenny expression actually block IMD signaling.

Major Points

1. In Figure 4C, ird5 distribution still seems diffuse, instead of the punctate state they showed in Figure 4b. This data does not fully support their interpretation, and is confusing at best.
2. In Figure 6, a Relish mutant control is absolutely necessary to demonstrate that the Relish specific antibody actually works as expected. This is not a widely used/validate reagent.
3. In Figure 6a, inclusion of a fly infection control is important, so the reader can see how this genetic manipulation compares to bona fide immune response.
4. Does Relish nuclear translocation also happen in gut, when autophagy is blocked?
5. In Figure 5C, the Kenny LIR domain mutant to show direct interaction between ird5 and Kenny is not inhibited due to LIR domain mutation.

Minor Points

1. In figure 1e, Y-Axis label is missing.
2. In figure 1C,D, they didn't show red and green channels except the merged data.
3. In figure 2d-I, they should have put the name of DAPI or nucleus in their images.

Reviewer #2 (Drosophila IKK, NFkB)(Remarks to the Author):

Tusco et al "Kenny mediates selective autophagic degradation of the IKK complex to control the innate immune response"

This manuscript identifies an important role of Kenny, the *Drosophila* IKK γ /NEMO protein, in the regulation of the anti-bacteria signaling pathway. Specifically, the authors report that Kenny interacts with ATG8a and IKK β /ird5, and is critically involved in the autophagic degradation of IKK β /ird5 during infection by serving as an autophagy receptor. The authors used a number of biochemical and genetic tools to establish that defects of autophagy in fruit flies (Atg8a and Atg7 mutants) lead to a fatal phenotype due to the presence of commensal microbiota. This is apparently caused by uncontrolled NF κ B signaling, as the elevated expression level of IMD target genes was observed in these mutants, and knocking-out Kenny prevents the prolonged expression of these target genes. While the results from *Drosophila* are solid, the authors could not detect an interaction between human IKK γ /NEMO and LC3/ GABARAP proteins. They thus developed a sophisticated mathematical model to show that the interactions between host and pathogens, the strength of innate immune response during infection, can drive the loss/gain of LIR (autophagy degradation signal) of IKK γ /NEMO in evolution.

Overall, this manuscript presents strong data in *Drosophila*, and clearly demonstrates the important role of Kenny in the termination of NF κ B signaling through selective autophagy pathway. I am not persuaded that the mathematical model is compelling, or appropriate for a Nature communications paper, as it does not rule out alternative explanations. It would seem more appropriate for a journal of theoretical evolution.

Specific questions/concerns:

1. The authors should establish either in cell culture or in flies, that the autophagy pathway, rather than the ubiquitin-proteasome pathway is the dominant mechanism for the turn-over of Kenny and IKK β /ird5 during infection.
2. Both the experimental data and the mathematical model strongly suggest the failure of Kenny-mediated IKK β /ird5 turn-over can lead to the fatal phenotype, it is critical for the author to demonstrate that flies with mutant Kenny (F7A/L10A, which abolishes the interaction between Kenny and ATG8a) also shows a similar fatal phenotype.
3. Data from *Drosophila* showed that Kenny is an autophagy receptor directly targeting IKK β /ird5 for degradation during bacterial infection, however, it is also possible that both Kenny and IKK β /ird5 can be targeted for degradation by other autophagy receptors. There are a number of selective autophagy receptors in mammals (for example, p62, OPTN, etc.), it is likely some of these host receptors can target NEMO/IKK β during NF κ B activation. Both NEMO and IKK β are ubiquitinated during NF κ B signaling and thus can readily be recognized as autophagy cargos. This should be included in the mathematical model as an additional component

Reviewer #3 (*Drosophila*, autophagy) (Remarks to the Author):

The manuscript by Tusco et al. describes the function of a LIR motif in Kenny, a necessary element of the *Drosophila* IMD signaling pathway, in mediating the autophagy mediated degradation of Kenny and the IRD5 component of the IKK complex. The work is original and illuminates an interesting connection between autophagy and elevated inflammatory signaling.

While the Kenny LIR motif does not appear to be conserved in mammalian homologs, the authors

offer an interesting model that attempts to place this difference in a physiological context.

There are however several weaknesses that the authors need to address:

1. A major weakness of the study is the reliance on overexpression of wt and mutant Kenny to assess the in vivo importance of the LIR motif. The dependence on autophagy as a degradative mechanism may be substantially exaggerated by the elevated protein levels.
2. The elevated levels of Kenny in autophagy mutants (Fig. 2b) appear to be largely of a different size than wt Kenny. The authors should test whether this represents a modified version of Kenny (e.g. ubiquitination, phosphorylation). The dependence of autophagic degradation of Kenny on such a modification would significantly alter the story.
3. From the data presented one would expect that Ird5 or mcherry-Ird5 would be more abundant in starved cells with Kenny knocked down by RNAi (Fig. 5g), but the opposite seems to be the case from the immunofluorescence images. Compare (Fig. 5f vs g). This should be tested by western blots.
4. To substantiate the claim that mammalian NEMO does not have functional LIR motifs – on which much of the mathematical modeling depends - the pulldowns in supplementary Fig. 7 need to be done with all six LC3/GABARAP homologs found in mammals in not just three of them. They are not all functionally equivalent.
5. The lack of conservation in mammals raises the question to which extend the Kenny LIR motif is conserved in other insects, as such conservation would further substantiate its importance.

Some minor details:

1. The co-ip from S2 cells in Fig. 1b does not look very convincing and should be quantified.
2. The Atg7 midgut image in Figure 2I does not match the controls (2J) and should be replaced
3. Units on the size measurement in Fig. 5h should be clarified
4. Tang et al. (PMID: 24268699) describe the connection between intestinal hyperplasia and the autophagy protein ATG9. This seems an important tangent that should be addressed in the discussion
5. Legend to Figure 2 “Full fly lysates” is a funny phrase, are these lysates from whole adult flies?
6. Figure 2 J to I. In the gut, in the absence of autophagy, Kenny is displaced from a beautiful perinuclear pattern to a punctate pattern. What are these punctae? They can't be the autophagosomal Kenny dots described in Figure 1.

Reviewers' comments:

Reviewer #1 (Remarks to the Author):

Tusco et al. in their model showed that autophagy controls IMD pathway by degrading Kenny and in autophagy mutants upregulation of IMD pathway is observed and that leads to gut hyperplasia. In our previous review, we noticed several weak points about their findings. This revised manuscript adequately deals with most of the earlier critiques, however some challenges remain;

1. The selectivity the degradation by autophagy pathway for Kenny remains unclear, and whether or not other IMD pathway components are also affected still has not be properly addressed. Although new data shows that ird5 is also degraded by autophagy, ird5 complexes tightly with Kenny and it is possible that ird5 and Kenny are degraded together. So, this results does not really address the specificity of this regulatory interaction. And this is an important issue, as the authors wish to claim that Kenny (&ird5) are the important cargo, in terms of the regulation of Imd signaling by autophagy. If other components are also targeted, these author target could explain their findings of increased signaling in autophagy mutants.

2. In response to the query about whether or not relish translocation is affected in autophagy mutants, they have shown that compared to wild type flies Atg8 mutant flies show relish translocation in the gut. However, the image they provided in the point by point response is surprising, because in their image in wild-type flies relish is not translocated in the nucleus at all, while several publications (Ryu et al.2008, Chen et al.2014) showed relish translocation in the gut of wild-type flies.

Reviewer #3 (Remarks to the Author):

In their revised version, the authors have adequately addressed most of the issues the reviewers have raised.

Somewhat disappointingly they decided to ignore my main concern whether the autophagic degradation they observe for Kenny is an artefact of its overexpression by the Gal4/UAS system or whether it also applies to Kenny expressed at endogenous levels. The experiment offered in response, overexpressed human GFP-NEMO, certainly does not address this concern.

I understand, however, that the generation of a tagged genomic Kenny transgene would a substantial burden, and its analysis would significantly delay publication of this otherwise very interesting and well-done paper. At the very least, this concern should be adequately acknowledged in the discussion.

REVIEWERS' COMMENTS:

Reviewer #1 (Remarks to the Author):

All concerns have been adequately addressed

Reviewer #3 (Remarks to the Author):

The authors have now fully addressed the concerns previously raised by the reviewers.

Response to reviewers

Manuscript NCOMMS-16-29645-T

We are grateful to the reviewers, as we believe their thoughtful constructive comments have helped us to substantially improve our manuscript.

Reviewer #1:

Tusco et al. propose an interesting model, where they argue that in flies, Kenny (the fly IKK γ /NEMO homolog) is selectively degraded by autophagy and this phenomenon is facilitated by the presence of AIM (Atg8 –Interacting-Motif) domain in Kenny. Furthermore, in autophagy mutants, up regulation of IMD pathway is observed and this results in the hyperplasia of gut, which ultimately leads to decreased fly survival.

Response: We appreciate the Reviewer's positive comments and interest in our work.

Major Concerns

Although their findings are fascinating, there are some weaknesses and poorly defined experiments that detract from the conclusions. For example, they claim that up regulation of the IMD pathway in the gut causes hyperplasia, but do not show that IMD signaling is increased specifically in this tissue.

Response: To answer reviewer's comment, we have examined the expression of *dipteracin* in isolated adult guts from Atg8a and Atg7 mutant flies by qPCR. We observed a very modest upregulation in isolated autophagy mutant guts (compared to the massive upregulation observed in full body extracts) (Supplementary Figure 10). This suggests that there is a systemic response regulated by other tissues. Indeed, we were able to show that expression of mCherry-Atg8a in the fat body was able to partially rescue upregulation of *dipteracin* in Atg8a mutant flies (Supplementary Figure 9). These results suggest that upregulation of AMPs in autophagy mutant guts is predominantly controlled by the fat body.

Moreover, the experimental data fails to demonstrate the IMD signaling is causative to the cut hyperplasia in the autophagy mutants. In another claim, they argue that autophagy selectively degrades Kenny happens, but didn't determine if other IMD pathway components are also degraded.

Response: To answer reviewer's comments, we have performed immunofluorescence analysis of phospho-Histone-3 (PH3) marker in Atg8a/Kenny mutant flies. Five days old adults were dissected and midguts were probed with 1:1000 of anti-pH3 antibody. Tissue was permeabilised with 0.1% Triton-X100 and 0.3% BSA, dissolved in PBS. We clearly show that there is no increased pH3 staining in Atg8a/Kenny mutants compared to Atg8a mutants, indicating that Kenny-mediated upregulation of AMPs is causative of gut hyperplasia in autophagy mutants (Supplementary Figure 11).

We have clearly shown that IMD pathway component *ird5* is degraded by autophagy (Figures 6 and 7).

Just how selective this degradation is, or what are the important targets in terms of regulating IMD signaling are not clear; for example Figure 6C appears to show elevated Relish in autophagy mutants. In particular, it seems far-fetched that Kenny persistence alone, due to autophagy inhibition, would drive AMP expression, as high levels of Kenny expression actually block IMD signaling.

Our results show that Kenny and *ird5* are degraded by autophagy and are accumulated when autophagy is inhibited. These increased endogenous levels of Kenny and *ird5* (not

overexpression - as correctly mentioned that has a dominant negative effect on IMD signaling) could drive AMP expression as we have shown in this manuscript.

Major Points

1. In Figure 4C, *ird5* distribution still seems diffuse, instead of the punctate state they showed in Figure 4b. This data does not fully support their interpretation, and is confusing at best.

Response: We apologize for the bad picture. The diffuse staining in Figure 4C is background staining. We now replaced Figure 4C with a better image. Figure 4 is now Figure 6 (we have moved two figures from the supplementary material to the main figures).

2. In Figure 6, a Relish mutant control is absolutely necessary to demonstrate that the Relish specific antibody actually works as expected. This is not a widely used/validate reagent.

Response: The Relish antibody that we used has been already used in other papers (Chen et al 2014, Cell 159(4):829-43. PMID: 25417159). In Chen et al. paper the antibody appears in materials and methods section as ‘Antibodies #Abin111036’ where as in this manuscript we mention that it is ‘RayBiotech 130-10080’. We apologize for the confusion but it is exactly the same antibody but it appears that RayBiotech is the distributor of Antibodies Online in UK. We have used Relish E20 mutants to test the specificity of the antibody. As shown in the figure below there is a significant reduction in nuclear Relish staining in Relish E20 mutants compared to wild type flies.

3. In Figure 6a, inclusion of a fly infection control is important, so the reader can see how this genetic manipulation compares to bona fide immune response.

Response: We now provide an infection control as requested (new Figure 8a). This shows that upon natural infection of wild-type flies with *Erwinia carotovora carotovora 15*, the level of activation of the IMD response is either comparable or higher than what was observed in autophagy depleted flies. For this experiment, five days old wild-type flies were pre-starved for 4 hours in empty tubes, and then kept for 24 hours on Whatman paper, soaked with a suspension of OD100 *Erwinia carotovora carotovora 15* with 2.5% sucrose. Following treatment RNA were extracted from flies and synthesized into cDNA, using the same protocol and materials as the other qPCR experiments. This was then amplified using *diptericin* primers, used as a marker of IMD activation.

4. Does Relish nuclear translocation also happen in gut, when autophagy is blocked?

Despite the fact that we observed a very modest upregulation in isolated autophagy mutant guts (see also major concerns above - Supplementary Figure 10), we have observed nuclear translocation of Relish in the midgut of *Atg8a*-mutant flies as shown in the figure below. A report by Ertürk-Hasdemir *et al*, 2009 PNAS proposed that the IKK complex is required both for the cleavage and phosphorylation of Relish, the latter being responsible for the efficient upregulation of AMP genes. There is evidence however that Relish cleavage alone may be enough to upregulate the expression of *Diptericin* only (Wiklund *et al*, 2009, Dev. Compar. Immunol.) Regulation of the IMD pathway is tissue-specific with its various output molecules being favoured differently from one tissue to another (Myllymäki *et al*, 2009, J Immunol). Nuclear translocation of Relish in *Atg8a*-mutants may lead to some level of *Diptericin* expression, however in the absence of infection this may in fact favour the expression of the IMD pathway antagonist Pirk and prevent Relish phosphorylation, which would have an overall negative effect on the local production of AMPs (Kleino *et al*, 2008 J Immunol).

5. In Figure 5C, the Kenny LIR domain mutant to show direct interaction between ird5 and Kenny is not inhibited due to LIR domain mutation.

Response: We have clearly showed in Figure 7 that Kenny-LIR mutant colocalises with ird5. We have also clearly showed using in vivo coimmunoprecipitation (coIP) (Supplementary Figure 7) that Kenny-LIR mutant is in a same complex with ird5. The coIP was done by precipitating GFP-Kenny-WT, GFP-Kenny-LIRm or GFP alone from lysates concomitantly expressing mCherry-ird5 using an anti-GFP antibody. mCherry-ird5 could be detected in the coIP with both GFP-Kenny constructs, but not with the GFP alone, thus showing that the interaction is specific to Kenny, but does not dependent on the functionality of the LIR motif. As reviewer requested, we additionally now show in GST pull down that Kenny-LIR mutant interacts with ird5 (Figure 7C).

Minor Points

1. In figure 1e, Y-Axis label is missing.

Response: This is corrected.

2. In figure 1 C, D. they didn't show red and green channels except the merged data.

Response: This is corrected.

3. In figure 2d-l, they should have put the name of DAPI or nucleus in their images.

Response: This is corrected.

Reviewer #2:

This manuscript identifies an important role of Kenny, the *Drosophila* IKK γ /NEMO protein, in the regulation of the anti-bacteria signaling pathway. Specifically, the authors report that Kenny interacts with ATG8a and IKK β /ird5, and is critically involved in the autophagic degradation of IKK β /ird5 during infection by serving as an autophagy receptor. The authors used a number of biochemical and genetic tools to establish that defects of autophagy in fruit flies (Atg8a and Atg7 mutants) lead to a fatal phenotype due to the presence of commensal microbiota. This is apparently caused by uncontrolled

NF κ B signaling, as the elevated expression level of IMD target genes was observed in these mutants, and knocking-out Kenny prevents the prolonged expression of these target genes. While the results from *Drosophila* are solid, the authors could not detect an interaction between human IKK γ /NEMO and LC3/ GABARAP proteins. They thus developed a sophisticated mathematical model to show that the interactions between host and pathogens, the strength of innate immune response during infection, can drive the loss/gain of LIR (autophagy degradation signal) of IKK γ /NEMO in evolution.

Overall, this manuscript presents strong data in *Drosophila*, and clearly demonstrates the important role of Kenny in the termination of NF κ B signaling through selective autophagy pathway. I am not persuaded that the mathematical model is compelling, or appropriate for a Nature communications paper, as it does not rule out alternative explanations. It would seem more appropriate for a journal of theoretical evolution.

Response: We appreciate the Reviewer's positive comments and interest in our work.

Specific questions/concerns:

1. The authors should establish either in cell culture or in flies, that the autophagy pathway, rather than the ubiquitin-proteasome pathway is the dominate mechanism for the turn-over of Kenny and IKK β /ird5 during infection.

Response: We have examined the accumulation of Kenny and ird5 when the proteasome is blocked. To impair to proteasome function, we have fed wild-type or Cg>mCherry-ird5 expressing flies for 5-6 days with the widely used proteasome inhibitor Bortezomib/PS-341 (5-20 μ M vs vehicle only). When the proteasome is blocked, we found that there is a modest accumulation of Kenny compared to one observed in autophagy mutants (Supplementary figure 4). This shows that Kenny may also be degraded by the proteasome but the dominate mechanism for its turnover is the autophagy pathway. Additionally, as explained below in Reviewer's 3 comment 2 and Discussion part, it appears that Kenny is accumulated in its phosphorylated form when autophagy is blocked but not when the proteasome is inhibited. This suggests that the modest proteasomal degradation of Kenny is not related to its role in autophagy. We have also found that ird5 (overexpressed mCherry-ird5) does not accumulate upon proteasome inhibition suggesting that autophagy is the major degradation pathway for ird5 (Supplementary figure 6).

2. Both the experimental data and the mathematical model strongly suggest the failure of Kenny-mediated IKK β /ird5 turn-over can lead to the fatal phenotype, it is critical for the author to demonstrate that flies with mutant Kenny (F7A/L10A, which abolishes the interaction between Kenny and ATG8a) also shows a similar fatal phenotype.

Response: We tried to express GFP-Kenny-WT and GFP-Kenny-LIR mutant in a Kenny mutant background (key1 mutant) but we were not able to generate the desirable genotypes probably due to embryonic lethality. To work around this, we did a clonal expression of GFP-Kenny-WT and GFP-Kenny-LIR mutant in adult midgut and stained for PH3 to show gut hyperplasia that is linked to the fatal phenotype we described. As shown in the figure below we were able to detect PH3-positive cells (arrowheads) only in cells expressing GFP-Kenny-LIR mutant and not in cells expressing GFP-Kenny-WT.

3. Data from *Drosophila* showed that Kenny is an autophagy receptor directly targeting IKK β /ird5 for degradation during bacterial infection, however, it is also possible that both Kenny and IKK β /ird5 can be targeted for degradation by other autophagy receptors. There are a number of selective autophagy receptors in mammals (for example, p62, OPTN, etc.), it is likely some of these host receptors can target NEMO/IKK β during NF κ B activation. Both NEMO and IKK β are ubiquitinated during NF κ B signaling and thus can readily be recognized as autophagy cargos. This should be included in the mathematical model as an additional component

Response:

Thank you for this suggestion, following which we have extended the model. In the original model, we considered competition between 2 hypothetical host types, with or without a LIR motif on IKK γ . We identified different forms of selective pressure that might drive one or other of these host types to succeed in a population. In the updated version of the model, we consider 4 hypothetical host types. Host types 1 and 2 do not use ubiquitin tagging to regulate IKK γ during infection, whilst host types 3 and 4 do; host types 2 and 4 have LIR motifs on IKK γ whilst types 1 and 3 do not. The updated model is described in our new Methods and Results sections and our results are presented in the new Figure 9. Our major conclusion remains that pathogen selection can plausibly drive the loss of LIR from IKK γ . We suggest two distinct ways this can occur. One possibility is that there is a specific advantage to regulating IKK γ during infection using ubiquitin tagging only. The second possibility is that infection mortality is minimized when the pathogen provides a LIR motif (essentially, when the pathogen manipulates the host immune response using a LIR motif, to the advantage of both host and pathogen). We discuss the biological relevance of these two possibilities in updated sections of the Discussion.

Reviewer #3:

The manuscript by Tusco et al. describes the function of a LIR motif in Kenny, a necessary element of the *Drosophila* IMD signaling pathway, in mediating the autophagy mediated degradation of Kenny and the IRD5 component of the IKK complex. The work is original and illuminates an interesting connection between autophagy and elevated inflammatory signaling. While the Kenny LIR motif does not appear to be conserved in mammalian homologs, the authors offer an interesting model that attempts to place this difference in a physiological context.

Response: We appreciate the Reviewer's positive comments and interest in our work.

There are however several weaknesses that the authors need to address:

1. A major weakness of the study is the reliance on overexpression of wt and mutant Kenny to assess the in vivo importance of the LIR motif. The dependence on autophagy as a degradative mechanism may be substantially exaggerated by the elevated protein levels.

Response: We have shown the in vivo functionality of LIR motif in Kenny by creating transgenic flies that overexpress Kenny-WT-GFP and Kenny -LIR mutant GFP (Figures 2,

7, S1 and S7). To further strengthen the *in vivo* functional importance of the LIR motif in Kenny, we have created transgenic flies that express GFP-HumanNEMO. We show that the lack of a functional LIR motif in NEMO results in the absence of colocalization between mCherry-Atg8a and GFP-NEMO (Supplementary Figure 15). This result further supports that LIR motif in *Drosophila* Kenny is crucial for its autophagic degradation.

2. The elevated levels of Kenny in autophagy mutants (Fig. 2b) appear to be largely of a different size than wt Kenny. The authors should test whether this represents a modified version of Kenny (e.g. ubiquitination, phosphorylation). The dependence of autophagic degradation of Kenny on such a modification would significantly alter the story.

Response: We have tested whether Kenny is phosphorylated or ubiquitinated. We have performed a dephosphorylation assay on lysates from autophagy-deficient flies (Atg8a and Atg7 mutants). The dephosphorylation assay has been performed using 1 unit of CIP (calf intestinal alkaline phosphatase) per 10 μ g total proteins (+phosphatase samples), a control has been included that contains a phosphatases inhibitor cocktail (+phosphatase +inhibitor samples); both test and control samples were compared to a non-treated sample (-phosphatase samples). All the samples have been incubated at 37°C for 15min before stopping the reaction by adding Laemmli buffer and heating the samples for 5 minutes at 95°C. A pan-phospho-proteins antibody has been used to assess the efficiency of the phosphatase treatments. We observed that the upper band of the doublet detected with the anti-Kenny antibody disappeared after treatment with the phosphatase (+phosphatase samples), but not in the presence of the inhibitor (+phosphatase +inhibitor samples) (Supplementary Figure 5a). These results show that Kenny is phosphorylated. Indeed it has been shown in that phosphorylation of various autophagy receptors enhances their autophagic degradation (Wild et al., *Science*. 2011 333:228-33, Rogov et al., *Biochem J*. 2013 454(3):459-66).

We have also tested whether Kenny is ubiquitinated in autophagy-deficient flies. We have expressed constitutively GFP-Kenny-WT with a tubulin-GAL4 driver in wild-type or Atg8a mutant flies. Denaturated full body fly lysates were prepared by boiling the lysates (13 flies in 200 μ L of lysis buffer 2% SDS) for 7 minutes at 95°C before 10 times dilution in buffer without SDS in order to reach a concentration of SDS lower than 0.2%. GFP-Kenny was immunoprecipitated using an anti-GFP antibody on clear lysates overnight at 4°C. We used an anti-ubiquitinated proteins antibody (FK2) to detect ubiquitin chains in the inputs and IP. We observed an accumulation of ubiquitinated protein in the input from Atg8a mutant flies compare to wild-type flies, as expected, but no enrichment of ubiquitinated GFP-Kenny in the IP (Supplementary Figure 5b).

3. From the data presented one would expect that Ird5 or mcherry-Ird5 would be more abundant in starved cells with Kenny knocked down by RNAi (Fig. 5g), but the opposite seems to be the case from the immunofluorescence images. Compare (Fig. 5f vs g). This should be tested by western blots.

Response: We have performed western blotting on lysates from starved flies expressing mCherry-ird5 along with key RNAi or a control luciferase RNAi in the fat body. We did observe a moderate increase in mCherry-ird5 when key is knocked down by RNAi (Supplementary figure 8). Due to the lack of anti-ird5 antibody, we cannot investigate the effect of Kenny silencing on endogenous ird5 protein accumulation, but instead we have to

overexpress mCherry-ird5 that could minimized the effects observed.

4. To substantiate the claim that mammalian NEMO does not have functional LIR motifs – on which much of the mathematical modeling depends - the pulldowns in supplementary Fig. 7 need to be done with all six LC3/GABARAP homologs found in mammals in not just three of them. They are not all functionally equivalent.

Response: We have tested all the six Atg8 family proteins in mammals and have found that NEMO does not interact with any of them (Supplementary Figure 14).

5. The lack of conservation in mammals raises the question to which extend the Kenny LIR motif is conserved in other insects, as such conservation would further substantiate its importance. **Response: Using BLAST analysis we found that Kenny LIR motif is conserved in other *Drosophila* species, in the common house fly (*Musca domestica*), mosquito (*Aedes aegypti*), butterfly (*Papilio xythus*) and silk moth (*Bombyx mori*). This suggests that there is conservation between Diptera and Lepidoptera (Supplementary Figure 13).**

Some minor details:

1. The co-ip from S2 cells in Fig. 1b does not look very convincing and should be quantified.

Response: We have now quantified the co-IP displayed in Figure 2 (shown in text).

2. The Atg7 midgut image in Figure 2I does not match the controls (2J) and should be replaced

Response: We have replaced figure 2I with a superficial confocal micrograph to match the controls (New Figure 3I).

3. Units on the size measurement in Fig. 5h should be clarified

Response: This is corrected.

4. Tang et al. (PMID: 24268699) describe the connection between intestinal hyperplasia and the autophagy protein ATG9. This seems an important tangent that should be addressed in the discussion

Response: We are now discussing about intestinal dysplasia and ATG9 as the reviewer suggested.

5. Legend to Figure 2 “Full fly lysates” is a funny phrase, are these lysates from whole adult flies?

Response: We apologize for the mistake, this is now rephrased.

6. Figure 2 J to I. In the gut, in the absence of autophagy, Kenny is displaced from a beautiful perinuclear pattern to a punctate pattern. What are these punctae? They can't be the autophagosomal Kenny dots described in Figure 1.

Response: We have used triple immunostaining for Kenny, ubiquitin and Ref(2)P and we show that Kenny puncta are sequestosomes that contain ubiquitin and Ref(2)P (Supplementary Figure 3).

Reviewers' comments:

Reviewer #1 (Remarks to the Author):

Tusco et al. in their model showed that autophagy controls IMD pathway by degrading Kenny and in autophagy mutants upregulation of IMD pathway is observed and that leads to gut hyperplasia. In our previous review, we noticed several weak points about their findings. This revised manuscript adequately deals with most of the earlier critiques.

We thank the reviewer for these encouraging comments.

However some challenges remain. 1. The selectivity the degradation by autophagy pathway for Kenny remains unclear, and whether or not other IMD pathway components are also affected still has not be properly addressed. Although new data shows that ird5 is also degraded by autophagy, ird5 complexes tightly with Kenny and it is possible that ird5 and Kenny are degraded together. So, this results does not really address the specificity of this regulatory interaction. And this is an important issue, as the authors wish to claim that Kenny (&ird5) are the important cargo, in terms of the regulation of Imd signaling by autophagy. If other components are also targeted, these author target could explain their findings of increased signaling in autophagy mutants.

We thank the reviewer for the thorough review of our manuscript. Our study focus on Kenny and ird5 degradation by autophagy. In the revised manuscript we have clearly showed that Kenny is the selective autophagy receptor for the degradation of the IKK complex. Since Kenny and ird5 form the IKK complex, we do propose that Kenny and ird5 are degraded together. This means that Kenny is interacting with the autophagic machinery (through its LIR-dependent interaction with Atg8a) and also interacts with ird5 (cargo). So, selective autophagy receptor (Kenny) and the cargo (ird5) are both degraded by autophagy. This is specific since Kenny interacts with Atg8a via its LIR motif and mediates the autophagic degradation of ird5. We cannot exclude that other components of the IMD pathway are also degraded by autophagy through interaction with Atg8a or other autophagy related proteins. The examination of autophagic degradation of other IMD components is beyond the scope of the current manuscript. We have now modified the discussion part to acknowledge this concern as follows:

'In conclusion, we have shown that autophagy plays a critical role for the termination of innate immune signalling in response to commensal microbiota by degrading Kenny and ird5. We cannot exclude that other components of the IMD pathway are also degraded by autophagy through interaction with Atg8a or other autophagy related proteins. Further studies are awaited to clarify these questions. Our study highlights the physiological importance of selective autophagy in the innate immune response of metazoans, and demonstrates the plasticity of its participating regulators.'

[Confidential data was redacted here]

2. In response to the query about whether or not relish translocation is affected in autophagy mutants, they have shown that compared to wild type flies Atg8 mutant flies show relish translocation in the gut. However, the image they provided in the point by point response is surprising, because in their image in wild-type flies relish is not translocated in the nucleus at all, while several publications (Ryu et al.2008, Chen et al.2014) showed relish translocation in the gut of wild-type flies.

We thank the reviewer for the thorough review of our manuscript. We apologize for the bad picture showing the wild type guts. To provide a better representative image, we have now repeated Relish immunostaining in the gut of wild type flies using exactly the same experimental procedures (age of the flies, buffers and antibody concentration) used in Chen et al 2014, Cell 159(4):829-43 (PMID: 25417159) and we can observe Relish translocation in the gut of wild type flies (see figure below). We have also added references PMID 23685204, 26827889 and 18218863 in the discussion.

Reviewer #3 (Remarks to the Author):

In their revised version, the authors have adequately addressed most of the issues the reviewers have raised.

We thank the reviewer for this positive feedback.

Somewhat disappointingly they decided to ignore my main concern whether the autophagic degradation they observe for Kenny is an artefact of its overexpression by the Gal4/UAS system or whether it also applies to Kenny expressed at endogenous levels. The experiment offered in response, overexpressed human GFP-NEMO, certainly does not address this concern.

I understand, however, that the generation of a tagged genomic Kenny transgene would a substantial burden, and its analysis would significantly delay publication of this otherwise very interesting and well-done paper. At the very least, this concern should be adequately acknowledged in the discussion.

Response:

We appreciate the reviewer's comment and we apologize about the possible confusion about this concern that we didn't mean to ignore. As the reviewer suggested, we have now modified the discussion part as follows:

'Selective autophagic degradation of Kenny is mediated by its LIR motif. We have shown the importance of the LIR motif for its degradation using 4 different technical approaches: 1) Mutation of the LIR motif drastically reduces the co-immunoprecipitation between Kenny and Atg8a when expressed in S2R+ cells, 2) Mutation of Kenny LIR motif abrogate the direct interaction with Atg8a,

as demonstrated by GST pulldown experiments. Similarly, Kenny loses its interaction with Atg8a lacking a functional LIR docking site (LDS), 3) Using tandem-tagged Kenny expressed in HeLa cells, we observed that Kenny-WT is targeted to acidic compartment (autolysosomes) when the cells are under starved condition (mostly red dots), while its LIR mutated counterpart is not (yellow only dots) and 4) GFP-Kenny ability to localize to autophagosomes and autolysosomes is compromised when its LIR motif is mutated. Although we were not able to show that autophagic degradation of a genomic Kenny LIR mutant was compromised, we have clearly shown that endogenous Kenny protein was robustly accumulated in Atg8a and Atg7 mutant flies.'